



# Quantifying the snowmelt–albedo feedback at Neumayer Station, East Antarctica

**Constantijn L. Jakobs**[1], **Carleen H. Reijmer**[1], **Peter Kuipers Munneke**[1], **Gert König-Langlo**[2], **and Michiel R. van den Broeke**[1]

[1]Institute for Marine and Atmospheric Research Utrecht, Utrecht University, Utrecht, the Netherlands
[2]Alfred-Wegener-Institut Helmholtz-Zentrum für Polar- und Meeresforschung, Bremerhaven, Germany

**Correspondence:** Constantijn L. Jakobs (c.l.jakobs@uu.nl)

**Abstract.** We use 24 years (1992–2016) of high-quality meteorological observations at Neumayer Station, East Antarctica, to force a surface energy balance model. The modelled 24-year cumulative surface melt at Neumayer amounts to 1154 mm water equivalent (w.e.), with only a small uncertainty ($\pm 3$ mm w.e.) from random measurement errors. Results are more sensitive to the chosen value for the surface momentum roughness length and new snow density, yielding a range of 900–1220 mm w.e. Melt at Neumayer occurs only in the months November to February, with a summer average of 50 mm w.e. and large interannual variability ($\sigma = 42$ mm w.e.). This is a small value compared to an annual average (1992–2016) accumulation of $415 \pm 86$ mm w.e. Absorbed shortwave radiation is the dominant driver of temporal melt variability at Neumayer. To assess the importance of the snowmelt–albedo feedback we include and calibrate an albedo parameterisation in the surface energy balance model. We show that, without the snowmelt–albedo feedback, surface melt at Neumayer would be approximately 3 CE1 times weaker, demonstrating how important it is to correctly represent this feedback in model simulations of surface melt in Antarctica.

## 1 Introduction

The Antarctic ice sheet (AIS) contains more than 25 million km$^3$ of ice, sufficient to raise global mean sea level by almost 60 m if melted completely (Fretwell et al., 2013). Between 1992 and 2017, the AIS lost mass at an accelerated rate, contributing $7.6 \pm 3.9$ mm to global sea level (Shepherd et al., 2018). This mass loss is mainly observed in coastal West Antarctica and the Antarctic Peninsula (AP) and is caused by glaciers that accelerated after their buttressing ice shelves had thinned or disintegrated (Wouters et al., 2015; Turner et al., 2017). The interaction between meltwater and firn, the intermediate product between snow and glacier ice, is hypothesised to play an important role in ice shelf disintegration (Kuipers Munneke et al., 2014). If the firn layer contains enough air, as is the case for most of the AIS, meltwater can percolate downwards and refreeze (Ligtenberg et al., 2014). If the storage capacity of the firn layer is reduced, surface meltwater can flow laterally towards the ice shelf edge (Bell et al., 2017), be stored englacially (Lenaerts et al., 2017) or form ponds on the ice shelf surface (Kingslake et al., 2017). In all cases, meltwater can accumulate in crevasses, thereby increasing the hydrostatic pressure in the crevasse tip, causing it to penetrate farther down. When a crevasse reaches the bottom of the ice shelf or a basal crevasse, part of the ice shelf disintegrates, a process called hydrofracturing (Van der Veen, 2007). Hydrofracturing has been identified as a potential precursor of rapid loss of Antarctic ice, accelerating sea level rise (DeConto and Pollard, 2016). In combination with enhanced ocean swell under low sea-ice conditions (Massom et al., 2018), hydrofracturing likely caused the disintegration of the Larsen B ice shelf in the AP in 2002 (Rignot et al., 2004; Scambos et al., 2004). In July 2017, a large iceberg calved from the Larsen C ice shelf, but it is unclear whether this signifies a further southward progression of ice shelf destabilisation in the AP (Hogg and Gudmundsson, 2017).

Improving our predictive capabilities of future ice shelf stability, AIS mass loss and associated sea level rise thus requires a thorough understanding of the surface melt process on Antarctic ice shelves. In contrast to meltwater occurrence, which is readily observed from space (Picard et al., 2007; Tedesco, 2009; Luckman et al., 2014), observational estimates of surface melt rates on Antarctic ice shelves are rare; they have been obtained locally through explicit modelling of the surface energy balance (SEB) (Van den Broeke et al., 2010; Kuipers Munneke et al., 2012, 2018). In turn, these enabled continent-wide melt rate estimates using calibrated satellite products based on backscatter strength of radio waves (Trusel et al., 2013, 2015). These studies invariably demonstrate that, in most parts of Antarctica, melt is currently a weak and intermittent process. In this melt regime, the positive snowmelt–albedo feedback (SMAF) plays a decisive role: when snow melts, meltwater may refreeze in the cold snowpack, resulting in considerably larger grains ($\sim 1$ mm) than new snow or snow that has been subjected to only dry compaction ($\sim 0.1$ mm). Larger snow grains reduce backward scattering of photons into the snowpack, increasing the probability of absorption and reducing the surface albedo, especially in the near-infrared (Wiscombe and Warren, 1980; Gardner and Sharp, 2010). This further enhances absorption of solar radiation and melt. For pure, uncontaminated snow, the strength of the SMAF depends on multiple factors, e.g. the intensity and duration of the melt and the frequency and intensity of snowfall events, which provide new snow consisting of smaller grains. We therefore expect the SMAF to be spatially and temporally variable on Antarctic ice shelves.

Most studies on the SMAF address the removal of (seasonal) snow and the appearance of dark soil or water (Perovich et al., 2002; Hall, 2004; Flanner et al., 2007; Qu and Hall, 2007), leading to further warming of the air and water. These studies commonly express the melt–albedo feedback in terms of air and water temperature sensitivity. Our aim is to quantify the impact on the melt rate of the darkening but not the disappearance of snow, a process addressed by far fewer studies (Box et al., 2012; Van As et al., 2013). To that end, we implement a snow albedo parameterisation (Gardner and Sharp, 2010; Kuipers Munneke et al., 2011b) in an SEB model, which is then calibrated using observations and used to study the sensitivity of melt rates to snow properties that influence snow albedo. We use 24 years of high-quality in situ observations (König-Langlo, 2017) from the German research station Neumayer (Fig. 1) to calculate the SEB and melt rate. We investigate the effects of measurement uncertainties and model settings on the modelled cumulative amount of surface melt. We then analyse the main drivers of surface melt and the magnitude of the SMAF at Neumayer by switching the feedback process in the albedo parameterisation on and off.

The SEB model is explained in Sect. 2.1, followed by a description of the albedo parameterisation in Sect. 2.2. The me-

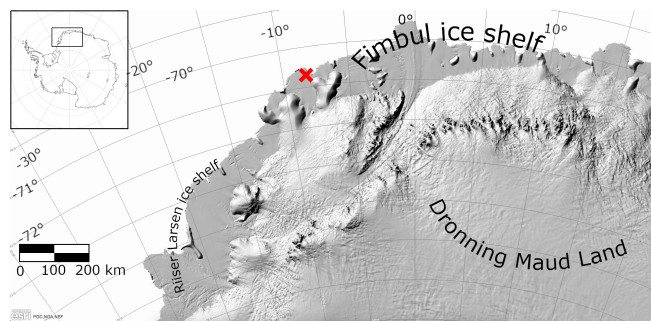

**Figure 1.** Map of the Antarctic continent. The red cross indicates the location of Neumayer Station. Imagery (©) 2016 DigitalGlobe, Inc.

teorological data used to force the SEB model are described in Sect. 2.3. The results section is split into two parts: in Sect. 3 we present and discuss the SEB and melt rate that are obtained using the observed albedo. In Sect. 4 the albedo parameterisation is used instead and the SMAF is quantified and discussed. Finally, the results are discussed in Sect. 5.

## 2  Methods

### 2.1  Surface energy balance model

The one-dimensional energy balance model is a further development of the models presented by Reijmer et al. (1999), Reijmer and Oerlemans (2002), Van den Broeke et al. (2005) and Kuipers Munneke et al. (2012); here only the main features are described. The energy balance of an infinitesimally thin surface layer (the "skin" layer) is defined as follows:

$$M = \mathrm{SW} \downarrow + \mathrm{SW} \uparrow + \mathrm{LW} \downarrow + \mathrm{LW} \uparrow + Q_S + Q_L + Q_G, \quad (1)$$

where positive fluxes are defined to be directed towards the surface. $\mathrm{SW} \downarrow$ and $\mathrm{SW} \uparrow$ are the incoming and reflected shortwave radiation, $\mathrm{LW} \downarrow$ and $\mathrm{LW} \uparrow$ are the downward and upward longwave radiation, $Q_S$ and $Q_L$ the turbulent sensible and latent heat fluxes and $Q_G$ is the conductive subsurface heat flux. We neglect latent energy from rain. $M$ is the energy used to melt snow or ice and is non-zero only when the surface has reached the melting point of ice ($T_s = 273.15$ K). Throughout this paper, melt and accumulation amounts are expressed in terms of millimetre water equivalent (mm w.e.), which equals kg m$^{-2}$. In order to calculate $Q_G$ and allow for densification, meltwater percolation and refreezing, a snow–firn model is used, initialised with 70 layers. The layer thickness varies from 1 cm at the top to 2 m at the bottom (25 m depth). We impose a no-energy flux boundary condition at the lowermost model level. New snow density is parameterised following the expression of Lenaerts et al. (2012), which relates it to the prevailing surface temperature ($T_s$) and 10 m wind speed ($V_{10\,\mathrm{m}}$) and imposes a lower limit of new snow density $\rho_{s,0}$. Meltwater per-

colation is based on the tipping-bucket method (e.g. Ligtenberg et al., 2011), allowing for immediate downward transport (within a single timestep of 10 s) of remaining water if a layer has attained its maximum capillary retention, as modelled using the expressions of Schneider and Jansson (2004). Meltwater refreezing increases the density and temperature of a layer. At the bottom of the firn layer, the meltwater is assumed to run off immediately, i.e. the model does not allow for slush/superimposed ice formation or lateral water movement. Turbulent fluxes are calculated following the "bulk" method, which is based on Monin–Obukhov similarity theory (see e.g. Van den Broeke et al., 2006 for relevant equations) between a single measurement level (2 m for temperature and humidity, 10 m for wind) and the surface, assuming the latter to be saturated with respect to ice and using the stability functions according to Dyer (1974) for unstable and Holtslag and De Bruin (1988) for stable conditions.

Subsurface penetration of shortwave radiation is calculated using a spectral model (Kuipers Munneke et al., 2009), based on the parameterisation by Brandt and Warren (1993), which is in turn based on the two-stream radiation model of Schlatter (1972). The impact on modelled melt and the quantification of the SMAF is discussed in the relevant sections.

The terms in Eq. (1) are either based on observations or can be expressed as a function of the skin temperature $T_s$. The SEB is solved iteratively by looking for a value of $T_s$ that closes the SEB to within 0.005 K between iterations: if $T_s > 273.15$ K, it is reset to 273.15 K and excess energy $M$ is used for surface melt. To evaluate model performance, the modelled value of $T_s$ is compared to observed $T_s$ calculated from LW $\uparrow$, using Stefan–Boltzmann's law for a longwave emissivity $\epsilon = 1$:

$$\text{LW} \uparrow = \sigma T_s^4, \tag{2}$$

where $\sigma = 5.67 \cdot 10^{-8}$ W m$^{-2}$ K$^{-4}$ is the Stefan–Boltzmann constant.

Surface roughness lengths for momentum, heat and moisture are related through the expression of Andreas (1987):

$$\ln\left(\frac{z_{0*}}{z_{0,m}}\right) = a_1 + a_2 \ln(Re_*) + a_3 \ln(Re_*)^2, \tag{3}$$

where $z_{0*}$ represents either $z_{0,h}$ or $z_{0,q}$, the roughness lengths for heat and moisture respectively, and $a_1$, $a_2$ and $a_3$ are coefficients determined by Andreas (1987) for various regimes of the roughness Reynolds number $Re_* = \frac{u_* z_{0,m}}{\nu}$ with kinematic viscosity $\nu$ and friction velocity $u_*$.

## 2.2 Albedo parameterisation

Because the shortwave radiation sensor faces the sky and includes a significant direct component, measured SW $\downarrow$ suffers from relatively large uncertainties owing to poor sensor cosine response, sensor tilt and/or rime formation (Smeets et al., 2018). In order to improve the accuracy of observed net

shortwave radiation used in the SEB calculations (Sect. 3), we calculate SW$_{\text{net}}$ based on SW $\uparrow$, which is diffuse and hence much less sensitive to these errors. To further decrease the impact of these errors, we use a 24 h moving average albedo, as described in Van den Broeke et al. (2004). In Sect. 4, in which albedo is parameterised to study melt–albedo feedbacks, for consistency we use measured SW $\uparrow$ in combination with parameterised albedo to estimate SW$_{\text{net}}$.

In Sect. 4, the parameterised surface albedo $\alpha$ is described as a base albedo $\alpha_S$, modified by perturbations describing the effect of changing solar zenith angle $\theta$ (d$\alpha_u$), the cloud optical thickness $\tau$ (d$\alpha_\tau$) and the concentration of black carbon in the snow (d$\alpha_c$) (Gardner and Sharp, 2010; Kuipers Munneke et al., 2011b):

$$\alpha = \alpha_S + d\alpha_u + d\alpha_\tau + d\alpha_c. \tag{4}$$

For Antarctica, we neglect the impact of impurities in the snow (d$\alpha_c = 0$); d$\alpha_u$ and d$\alpha_\tau$ both depend on the base albedo $\alpha_S$, d$\alpha_u$ in addition depends on the solar zenith angle ($u = \cos\theta$) and d$\alpha_\tau$ on the cloud optical thickness $\tau$:

$$d\alpha_u = 0.53\alpha_S(1 - \alpha_S)(1 - 0.64x - (1-x)u)^{1.2}, \tag{5}$$

$$d\alpha_\tau = \frac{0.1\tau(\alpha_S + d\alpha_c)^{1.3}}{(1 + 1.5\tau)^{\alpha_S}}, \tag{6}$$

where $x = \min\left(\frac{\sqrt{\tau}}{3u}, 1\right)$. The base albedo depends on the snow grain size $r_e$ (in metres):

$$\alpha_S = 1.48 - 1.27048 r_e^{0.07}, \tag{7}$$

in which the snow grain size $r_e$ on time step $t$ is parameterised as

$$r_e(t) = [r_e(t-1) + dr_{e,\text{dry}} + dr_{e,\text{wet}}]f_o + r_{e,0}f_n + r_{e,r}f_r. \tag{8}$$

Here, $dr_{e,\text{dry}}$ and $dr_{e,\text{wet}}$ describe the metamorphism of dry and wet snow respectively, $f_o$, $f_n$ and $f_r$ are the fractions of old, new and refrozen snow, and $r_{e,0}$ and $r_{e,r}$ are the grain sizes of new and refrozen snow. Dry snow metamorphism is parameterised following Kuipers Munneke et al. (2011b):

$$\frac{dr_{e,\text{dry}}}{dt} = \left(\frac{dr_e}{dt}\right)_0 \left(\frac{\eta}{(r_e - r_{e,0}) + \eta}\right)^{1/\kappa}, \tag{9}$$

where $r_{e,0}$ is the new snow grain size, and the coefficients $\left(\frac{dr_e}{dt}\right)_0$, $\eta$ and $\kappa$ are obtained from a look-up table. This look-up table is compiled based on simulations with the SNICAR model (Flanner and Zender, 2006), which calculates the snow metamorphism resulting from temperature gradient metamorphism. $dr_{e,\text{wet}}$ is a function of the snow grain size $r_e$ itself and the liquid water content $f_{\text{liq}}$ (Brun et al., 1989):

$$\frac{dr_{e,\text{wet}}}{dt} = \frac{C f_{\text{liq}}^3}{4\pi r_e^2}, \tag{10}$$

where $C$ is a constant ($4.22 \cdot 10^{-13} \text{ m}^3 \text{ s}^{-1}$).

The fractions $f_o$, $f_n$ and $f_r$ are derived from the snow/firn model, and the grain sizes of new and refrozen snow are constants; the method for determining their values from a tuning exercise is described in Sect. 4.1.

To determine cloud optical thickness $\tau$, an empirical relation between $\tau$ and the longwave-equivalent cloud cover $N_\epsilon$ is used following Kuipers Munneke et al. (2011a):

$$\tau = c_1 \left( \exp(c_2 N_\epsilon) - 1 \right), \tag{11}$$

with fitting parameters $c_1$ and $c_2$. $N_\epsilon$ is determined using a method described by Kuipers Munneke et al. (2011a), which relates hourly values of downward longwave radiation LW ↓ to near-surface air temperature $T_{2\,\text{m}}$ as illustrated in Fig. 2a. Red lines indicate quadratic fits through the upper and lower 5 percentiles of the data, assumed to represent fully cloudy and clear conditions, respectively. $N_\epsilon$ is obtained by linearly interpolating between these upper and lower bounds, yielding values between 0 and 1. Hourly values for cloud cover are then used to obtain values for $\tau$ (Fig. 2b). The values used for the fit parameters $c_1 = 5.404$ and $c_2 = 2.207$ (both dimensionless) differ somewhat from Kuipers Munneke et al. (2011a), who used daily values for the fit.

## 2.3 Observational data

The SEB model is forced with data from the meteorological observatory at the German research station Neumayer, situated on the Ekström ice shelf (König-Langlo, 2017). The observatory has been operational since 1981 and was relocated in 1992 and 2009. In 2016, its location was 70°40′ S, 8°16′ W (Fig. 1). The observatory is one of only four Antarctic stations – and the only one situated on an ice shelf – that is part of the Baseline Surface Radiation Network (BSRN), a global network of stations with high-quality (artificially ventilated) radiation observations, coordinated by the Alfred Wegener Institute (AWI). We use hourly averages of 2 m temperature ($T_{2\,\text{m}}$) and specific humidity ($q_{2\,\text{m}}$), 10 m wind speed ($V_{10\,\text{m}}$), surface pressure ($p$) and radiation fluxes for the period April 1992–January 2016 (24 years) to force the SEB model; their uncertainty ranges are provided in Table 1. Approximately 4.1 % of the data points contained at least one missing variable, which mostly come from daily performed visual inspection of the data. To obtain a continuous data set, all missing data were replaced: pressure, relative humidity, wind speed, temperature and longwave radiation were simply linearly interpolated. In the case of shortwave radiation, the missing value was replaced by imitating the average daily cycle of the 2 preceding days. As the measurement station is visited and maintained every day, the impact of rime formation is limited, as is the tilt of the observation mast, resulting in a high-quality meteorological data set.

Accumulation observations are only available from stake measurements, provided by AWI, which were performed weekly for the period April 1992–January 2009. As timing

**Table 1.** Listing of used measurement variables and their associated measurement uncertainties.

| Variable | Uncertainty range |
|---|---|
| $V_{10\,\text{m}}$ | max ($0.5 \text{ m s}^{-1}$, 5 %) |
| SW ↓ | $5 \text{ W m}^{-2}$ |
| SW ↑ | $5 \text{ W m}^{-2}$ |
| LW ↓ | $5 \text{ W m}^{-2}$ |
| LW ↑ | $5 \text{ W m}^{-2}$ |
| $T_{2\,\text{m}}$ | 0.1 °C |
| $RH_{2\,\text{m}}$ | 5 % |
| $p$ | 0.5 hPa |

of precipitation is important for correctly simulating the effects of new snow on snow albedo, we combined the stake observations with precipitation predicted by the regional atmospheric climate model RACMO2.3p2 (Van Wessem et al., 2018) to obtain realistic timing of precipitation in between stake observations, as well as for the post-2009 period. The amount of precipitation modelled by RACMO2 was scaled such that the modelled surface height changes agree with stake measurements; this required a 15.3 % upward adjustment of the modelled precipitation flux.

### Local near-surface climate

Neumayer station is located on an ice shelf $\sim 20 \text{ km}$ from Halvfarryggen ice rise to the south-east, $\sim 100 \text{ km}$ from the ice shelf break (grounding line) to the south, $\sim 20 \text{ km}$ from open water and sea ice to the north and $\sim 5 \text{ km}$ to open water and sea ice to the east. As a result, Neumayer experiences relatively mild conditions without significant impact from katabatic winds but with a pronounced influence of synoptic low-pressure systems passing mainly from west to east in the South Atlantic Ocean to the north of the station. The seasonal cycles of 2 m temperature, 10 m wind and 2 m specific humidity are presented in Fig. 3a. Summer temperatures around $-4$ °C and winter temperatures around $-25$ °C imply a substantial ($> 20$ K) seasonal temperature amplitude based on monthly mean values. This is in line with the formation of a surface-based temperature inversion in winter, a phenomenon that is representative for the flat ice shelves as well as the interior ice domes and in contrast to the topographically steeper escarpment zone, where the quasi-continuous mixing by katabatic flow limits the formation of such an inversion (Van den Broeke, 1998). As expected from the strong link to the air temperature through the Clausius–Clapeyron relation and a high annual mean relative humidity of 82 % (relative to either water or ice, depending on the air temperature), because of the proximity of a saturated snow surface and the ocean, the seasonal cycle of $q_{2\,\text{m}}$ closely follows that of temperature.

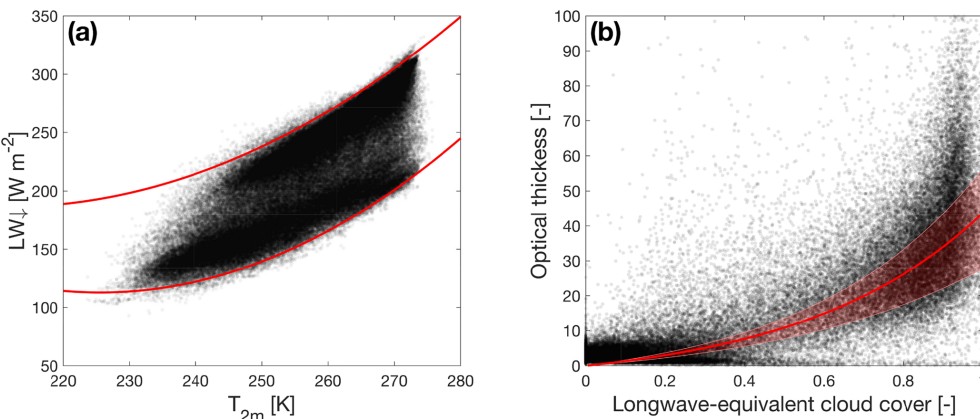

**Figure 2. (a)** Downward longwave radiation vs. air temperature. The red lines are quadratic fits of the upper and lower 5 percentile boundaries. The longwave-equivalent cloud cover is determined by linear interpolation between these bounds. **(b)** Optical thickness vs. cloud cover. The red line resembles the best fit to a function $\tau = c_1 \left( e^{c_2 N_\epsilon} - 1 \right)$. The shaded area indicates the 95 % uncertainty range.

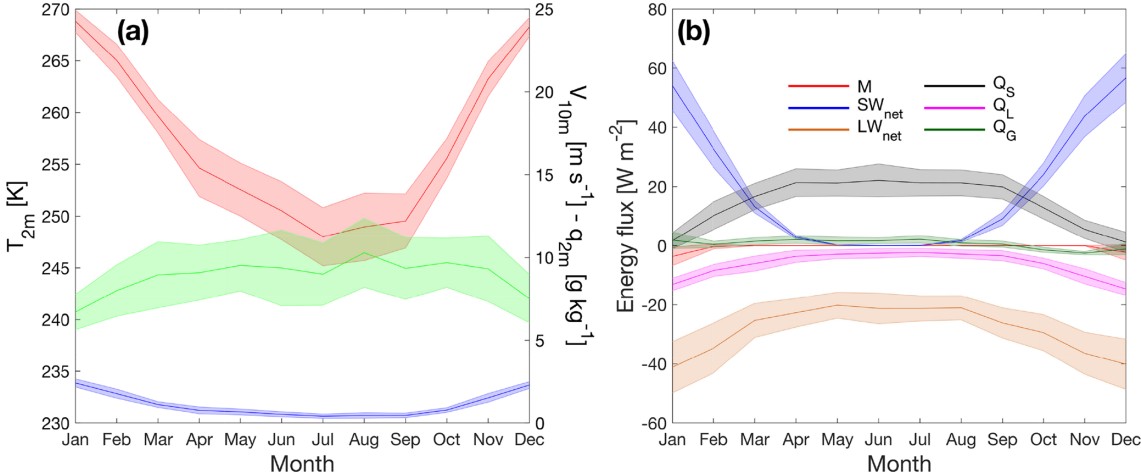

**Figure 3. (a)** Seasonal cycles of 2 m temperature (red, left axis), 10 m wind speed (green, right axis) and 2 m specific humidity (blue, right axis). Shaded areas indicate the standard deviations of monthly means. **(b)** Same as panel **(a)** for melt (red), net shortwave radiation (blue), net longwave radiation (orange), sensible heat (black), latent heat (magenta) and ground heat (green).

## 3 Results: surface energy balance and melt

### 3.1 SEB model performance and uncertainties

There are several SEB model parameters for which the exact values or formulations are unknown, e.g. the surface roughness length for momentum $z_{0,m}$, the density of new snow $\rho_s$, the stability functions (required to calculate the turbulent scales) and the effective conductivity, which couples the magnitude of $Q_G$ to the temperature gradient in the snow. We estimated the impact of observational and model uncertainties on modelled melt by running the model 600 times while randomly varying all hourly observations within the specified measurement uncertainty ranges (Table 1) and using multiple expressions for the heat conductivity and stability functions. Model performance is quantified by compar-

ing modelled with observed $T_s$ and assessing the changes in modelled 24-year cumulative melt. Note that in this section, the albedo based on observations is used to obtain $SW_{net}$.

The choice of expressions for the stability functions and heat conductivity did not significantly impact the modelled amount of melt (total within 30 mm w.e. or 2.7 %, not shown). The model outcome is more sensitive to the choice of surface roughness length for momentum $z_{0,m}$ and the lower limit of density of new snow $\rho_{s,0}$: when $z_{0,m}$ is varied between 0.5 and 50 mm and $\rho_{s,0}$ between 150 and 500 kg m$^{-3}$, the cumulative amount of surface melt over the 24-year period varies between 900 and 1220 mm w.e., with higher melt values for smaller values of $z_{0,m}$ and $\rho_{s,0}$. Optimal values in terms of simulated $T_s$ are $z_{0,m} = 1.65$ mm and $\rho_{s,0} = 280$ kg m$^{-3}$, resulting in a $T_s$ bias of 0.01 K and an RMSD of 0.79 K (Fig. 4). We use these values in the

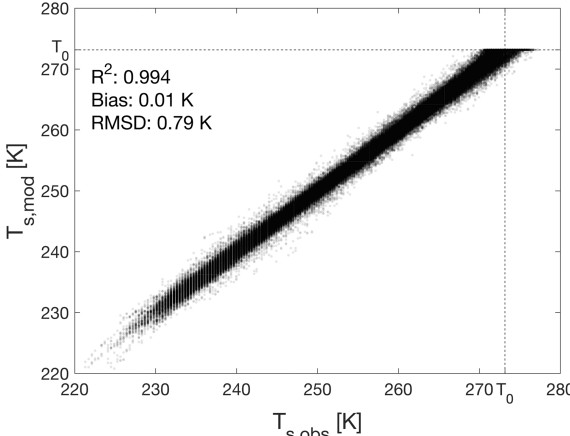

**Figure 4.** Daily values of modelled vs. measured $T_s$ for the parameter settings used in this study: $z_{0,m} = 1.65\,\mathrm{mm}$, $\rho_{s,0} = 280\,\mathrm{kg\,m^{-3}}$.

remainder of this study. Figure 5a and b show modelled 24-year cumulative melt and annual melt (March–February) at Neumayer, combined with uncertainties associated with model parameters. The annual mean values for year $X$ are obtained by averaging monthly values for March of year $X$ until February of year $X + 1$ TS1. The total melt amounts to 1154 mm w.e., with a small uncertainty associated with measurement uncertainties ($1\sigma \approx 3$ mm w.e., i.e. 0.3 %). The method adopted to estimate this uncertainty has its limitations, as measurement errors are probably autocorrelated: if a measurement at one time is disturbed in some way, it is probably disturbed in a similar way at the next time step. Therefore, this result could be interpreted as a lower bound of the uncertainty range, which is supported by the larger uncertainty estimates ($\sim 15$ %) by Van den Broeke et al. (2010), who applied a constant systematic error which can be interpreted as an upper bound on the modelled uncertainty range. This also explains why the model outcome is much more sensitive to different values of $z_{0,m}$, as these runs effectively introduce a systematic error between the true (unknown) value and the chosen value. Furthermore, this approach assumes the true value to be constant, which likely is an oversimplification (Smeets and Van den Broeke, 2008).

The sensitivity of modelled cumulative melt to $z_{0,m}$ is somewhat unexpected. Following Eq. (3) both $z_{0,h}$ and $z_{0,q}$ decrease for increasing $z_{0,m}$; in combination with the bulk method this acts to dampen the effect of $z_{0,m}$ on the magnitude of the turbulent fluxes. Our interpretation of this result is that decreasing $z_{0,m}$ and $\rho_{s,0}$ lead to a decrease in the turbulent fluxes as well as the ground heat flux $Q_G$. This reduces the efficiency with which heat is removed from the surface, in turn allowing more energy to be invested in melt. The obtained value of $z_{0,m} = 1.65$ mm is high compared to the average value of $z_{0,m} = 0.1$ mm found during a field campaign at Neumayer in 1982 (König, 1985) but it is not uncommon for snow surfaces (Amory et al., 2017).

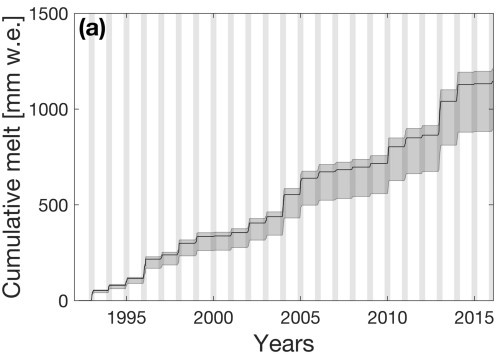

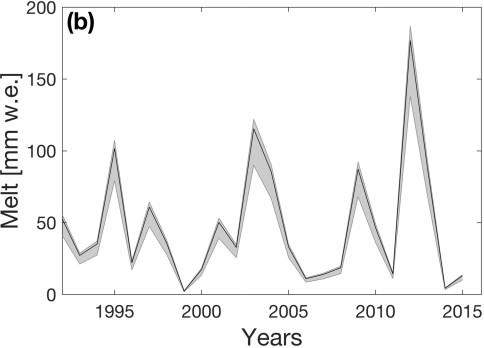

**Figure 5.** Effect of model uncertainties on **(a)** cumulative melt and **(b)** seasonal melt. The shaded area indicates the $1\sigma$ range due to model uncertainties (changing $z_{0,m}$ and $\rho_{s,0}$ between their respective values). The vertical grey patches in panel **(a)** indicate November–February of each season. Note that panel **(b)** ends earlier than panel **(a)** because the observations do not cover the 2015–2016 melt season entirely.

Measured values of $T_s$ in excess of the melting point in Fig. 4 only occurred in the first six seasons; from 1998–1999 onwards they were removed by additional post-processing. These measurements mainly reflect uncertainties in the adopted unit value of longwave emissivity and in measured LW $\uparrow$, e.g. from sensor window heating (Smeets et al., 2018) and the fact that the downward-facing radiation sensor also measures longwave radiation emitted by the relatively warm air between the surface and the sensor.

## 3.2 Surface energy balance

Annual (March–February) mean values of near-surface meteorological quantities and SEB components are presented in Table 2, with seasonal cycles of SEB components presented in Fig. 3b. These show that the summertime SEB is dominated by the radiation fluxes; despite the high albedo of the snow surface, $SW_{net}$ is the dominant heat source for the skin layer, whereas $LW_{net}$ extracts energy from the surface, most efficiently so in summer, when the surface is heated by the sun. In summer, $Q_L$ becomes a significant source of heat loss in the SEB (sublimation), preventing strong negative $Q_S$

**Table 2.** Mean annual values and interannual variability (calculated as standard deviations of monthly means) of meteorological variables and SEB components. For precipitation and melt, total annual values are given.

| Variable | Yearly mean | Variability |
|---|---|---|
| $T_{2\,m}$ (K) | 257.1 | 0.7 |
| $T_s$ (K) | 256.0 | 0.8 |
| $q_{2\,m}$ (g kg$^{-1}$) | 1.1 | 0.1 |
| $V_{10\,m}$ (m s$^{-1}$) | 8.9 | 0.6 |
| $p$ (hPa) | 981.6 | 2.0 |
| SW$_{net}$ (W m$^{-2}$) | 20 | 2 |
| SW ↓ (W m$^{-2}$) | 127 | 3 |
| SW ↑ (W m$^{-2}$) | 107 | 2 |
| LW$_{net}$ (W m$^{-2}$) | −28 | 3 |
| LW ↓ (W m$^{-2}$) | 218 | 5 |
| LW ↑ (W m$^{-2}$) | 246 | 4 |
| $Q_S$ (W m$^{-2}$) | 14.5 | 2.7 |
| $Q_L$ (W m$^{-2}$) | −6.3 | 1.2 |
| $Q_G$ (W m$^{-2}$) | 0.7 | 0.4 |
| $M$ (W m$^{-2}$) | 0.5 | 0.4 |
| Precipitation (mm w.e.) | 415 | 86 |
| Melt (mm w.e.) | 50 | 42 |

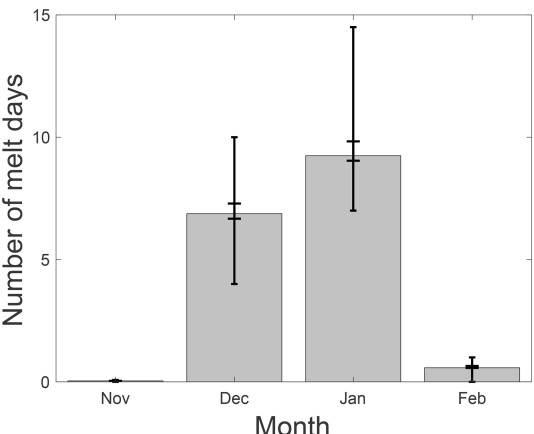

**Figure 6.** Average number of melt days per month at Neumayer. The inner error bars (with larger caps) indicate the $1\sigma$ uncertainty range resulting from the runs performed with different settings for roughness length $z_0$ and lower limit of new snow density $\rho_{s,0}$ (Sect. 3.1). The outer error bars (with smaller caps) indicate the $1\sigma$ range of the interannual variability.

(convection). The seasonal cycle of $Q_G$ is small, indicating a small net transport of heat away from the surface in summer and towards the surface in winter. The net annually integrated amount is less than zero as a result of the refreezing of meltwater, warming the subsurface snow layers.

Statistically significant and previously unreported trends over the full 24-year period (not shown) are detected in LW ↑ ($-0.28 \pm 0.14$ W m$^{-2}$ yr$^{-1}$) and $Q_S$ ($+0.21 \pm 0.07$ W m$^{-2}$ yr$^{-1}$). Both of these are a result of wintertime trends. LW ↑ is linked directly to $T_s$, which shows a statistically insignificant negative trend ($-0.029 \pm 0.026$ K yr$^{-1}$), which in magnitude exceeds the negative trend in $T_{2\,m}$ ($-0.0045 \pm 0.02$ K yr$^{-1}$; assuming a normal distribution, the probability that the negative trend in $T_s$ is greater in magnitude than the trend in $T_{2\,m}$ is 0.76). As a result, the air temperature gradient near the surface has increased, enhancing $Q_S$. The negative trend in $T_s$ originates from a decrease in LW ↓ ($-0.26 \pm 0.17$ W m$^{-2}$ yr$^{-1}$), which is in turn driven by a slight decrease in cloud cover ($-0.003 \pm 0.001$ yr$^{-1}$). This is suggested independently by the decrease in average winter humidity ($-0.004 \pm 0.002$ g kg$^{-1}$ yr$^{-1}$). These findings agree with Herman et al. (2013) and Kuipers Munneke et al. (2011a), who determined from satellite observations that summer cloud cover has decreased over that part of coastal Antarctica in the period 1979–2011.

### 3.3  Melt season

Melt occurs at Neumayer from November to February (Fig. 6) but is highly variable from year to year. The mean an-

nual amount of melt is 50 mm w.e. with an interannual variability of 42 mm w.e. and a range of 2 mm w.e. in 1999–2000 to 176 mm w.e. in 2012–2013. Most melt occurs in December and January and the surface only sporadically reaches melting point in February. Only in 2007 did melt occur in November, and no melt occurs outside these 4 months. The cumulative melt occurring at Neumayer shows stepwise increases (Fig. 5a), which represent the peaked melt seasons, in which melt occurs on average on $18 \pm 10$ d CE2. The uncertainty in the number of melt days due to the chosen values of $z_{0,m}$ and $\rho_{s,0}$ is relatively small compared to the interannual variability in melt totals (Fig. 6), implying that this choice does not significantly affect the modelled melt duration, but it does affect the total melt.

To investigate the link between melt and climate, we compare the two summers with the highest (2003–2004 and 2012–2013, on average 145 mm w.e.) and lowest (1999–2000 and 2014–2015, on average 4 mm w.e.) melt amounts. Figure 7 shows the meteorological and SEB components for these years, averaged over December and January. The largest differences are found in $T_{2\,m}$ ($+2.3$ K) and SW$_{net}$ ($+17$ W m$^{-2}$); based on the measurement uncertainties (Table 1), these differences are significant. In cold summers, the low $T_{2\,m}$ corresponds to a stronger temperature inversion ($T_{2\,m} - T_s$), more longwave cooling, less sublimation and a larger $Q_S$. SW ↓ and LW ↓ show almost no difference between high and low melt seasons; therefore, the difference in SW$_{net}$ cannot be caused by a change in cloud cover and is likely caused solely by surface albedo, which suggests an important role for the SMAF. This will be elaborated upon in the next section. Finally, the direction of $Q_G$ is reversed: in high melt years, the surface is warmed from below, while in low melt years the surface loses heat to the subsurface. More

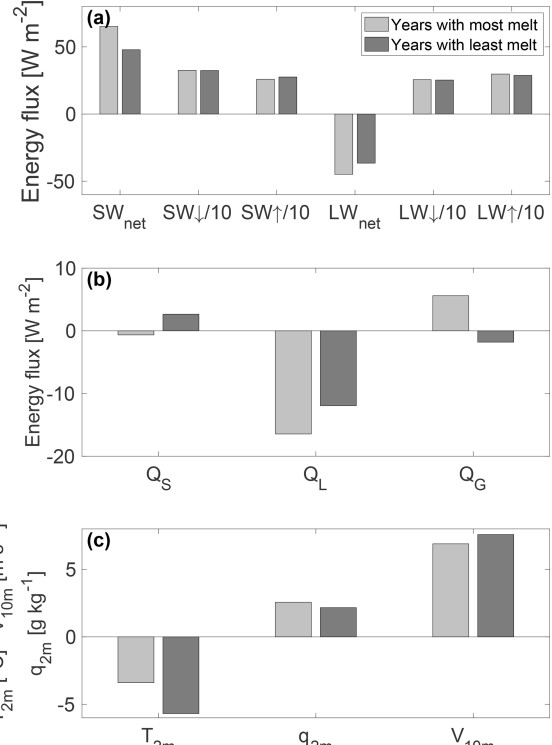

**Figure 7.** Average values of some SEB components **(a, b)** and some meteorological variables **(c)** for December and January in the years with the highest (2003–2004 and 2012–2013, in light grey) and lowest (1999–2000 and 2014–2015, in dark grey) amount of melt, as identified in Sect. 3.3. Note that SW ↓, SW ↑, LW ↓ and LW ↑ are scaled by a factor of 10 in panel **(a)** for clarification.

refreezing of meltwater in high melt years warms the near surface snow layers, which in turn leads to a conductive heat flux towards the surface.

Using the subsurface radiation model of Kuipers Munneke et al. (2009), the influence of subsurface penetration of short-wave radiation is estimated. Its inclusion increases the modelled cumulative amount of melt by 13 %, from 1154 to 1326 mm w.e. The absorbed shortwave radiation heats the subsurface layers, but the heat cannot be transported away as effectively as would happen at the surface by turbulent fluxes and longwave radiation. This leads to an increase in total melt.

The findings presented in this section are in good agreement with Van den Broeke et al. (2010), who used a similar approach to calculate the SEB at Neumayer but used a lower value for $z_{0,m} = 0.32$ mm and a higher snow density that was assumed constant with depth (420 kg m$^{-3}$ in their study vs. 280 kg m$^{-3}$ in this study). Compared to melt estimates from the Larsen C ice shelf, obtained through a similar modelling approach by Kuipers Munneke et al. (2012), melt at Neumayer is weak. Owing to its more northerly location, on the Larsen C ice shelf an annual (2009–2011) average melt energy of 2.8 W m$^{-2}$ is obtained, compared to the 2009–2011

annual average of 0.7 W m$^{-2}$ obtained at Neumayer. Furthermore, in November and February melt occurs much more frequently on the Larsen C ice shelf.

## 4 Results: the snowmelt–albedo feedback

The SMAF is a well-known phenomenon but has not before been quantified for Antarctica. The feedback occurs after the rapid growth of snow grains when meltwater penetrates into the subsurface and refreezes. Because a photon travels farther through snow with large particles than in new snow with smaller particles on average, the probability of it being absorbed is increased, effectively lowering the surface albedo (Gardner and Sharp, 2010). Even without melt, albedo decreases when snow ages, following grain growth from dry snow metamorphism, but this is a much slower process which mainly depends on temperature gradients in the snow, favouring moisture transport onto larger grains. Precipitation of new, fine-grained snow has been shown to inhibit the albedo decrease by metamorphism on the Antarctic plateau (Picard et al., 2012).

To quantify the SMAF at Neumayer, we need to be able to switch on and off the albedo dependency on melt-driven grain growth. To that end, we implemented an albedo parameterisation in the SEB model, as described in Sect. 2.2. Because no data on grain size are available from Neumayer, we optimised the albedo model performance by maximising the correspondence between (1) modelled and observed hourly SW ↑ and (2) the total melt obtained from the calculations based on observed albedo (Sect. 4.1). We compare SW ↑ instead of the albedo itself because by doing so the hourly values are naturally weighted with its contribution to SW$_{net}$ and hence its importance for the SEB. We then perform several runs with different processes switched on and off affecting the surface albedo to investigate the importance of the SMAF for melt at Neumayer (Sect. 4.2).

### 4.1 Optimising the albedo parameterisation

The albedo parameterisation, and especially the expression for snow grain size (Eq. 8), contains several parameters that are not well constrained, such as new snow grain size $r_{e,0}$ and refrozen snow grain size $r_{e,r}$. These parameters were varied within reasonable ranges to optimise the results: new snow grain sizes between 0.04 and 0.3 mm and refrozen snow grain sizes between 0.1 and 10 mm. The best comparison with observed albedo was achieved when using the look-up table for dry snow metamorphism, $dr_{e,dry}$, corresponding to a grain size of 0.055 mm.

The first step in optimising the parameterisation was to split the summer season into two parts, the "dry" and the "wet" season. The respective starts of the dry and wet seasons are the first day on which the sun rises more than 15° above the horizon and the first day that surface melt occurs. The wet

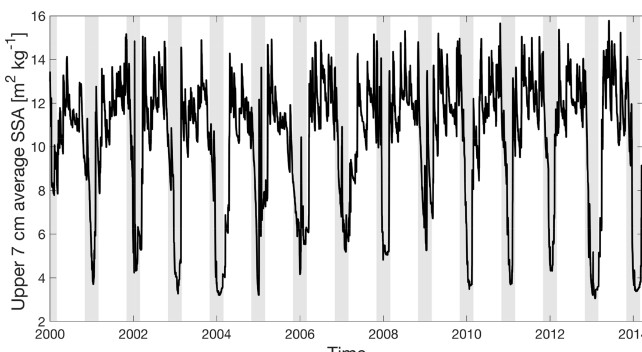

**Figure 8.** Seasonal cycle of modelled average grain size in the upper 7 cm for the period 2000–2014. The grain size is expressed in terms of specific surface area ($SSA = \frac{3}{\rho_i r_e}$) rather than grain size itself to allow for a comparison with Fig. 6 of Libois et al. (2015). The vertical grey patches indicate November–February of each season. TS2

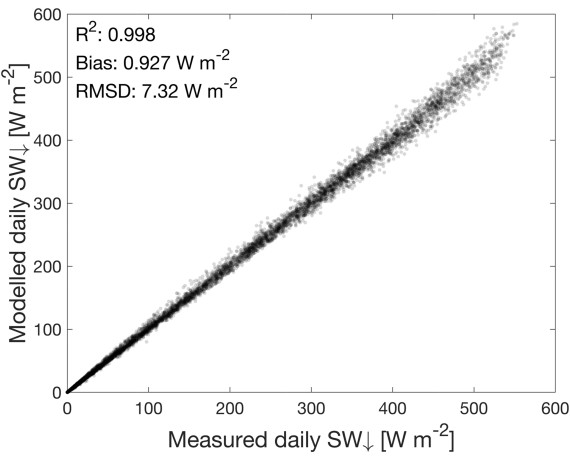

**Figure 9.** Measured vs. modelled daily average incoming shortwave radiation (SW ↓). The modelled SW ↓ is obtained by dividing the hourly measured SW ↑ by the hourly modelled albedo.

season ends when the sun no longer rises higher than 15°. For the dry season, we varied the dry snow metamorphism factor and the new snow grain size to best match observed SW ↑. This resulted in a new snow grain size of 0.25 mm. This value is then used in the second step, in which the refrozen snow grain size $r_{e,r}$ is varied to best match the modelled cumulative melt using observed albedo. This was achieved for a refrozen snow grain size of 1.45 mm.

This value for refrozen snow grain size is compatible with the typical largest grains in dry metamorphosed snow of O(1 mm) and which Kuipers Munneke et al. (2011b) used as a lower limit for refrozen snow grains. Libois et al. (2015) and Picard et al. (2016) present observations of snow grain sizes on the Antarctic plateau during field campaigns in 2012–2013 and 2013–2014 as well as estimates from satellite observations. On the plateau, summer temperatures are comparable to Neumayer winter temperatures. Libois et al. (2015) report summertime snow grain size estimates of approximately 0.11 mm (Fig. 6 in their study, reported as a specific surface area $SSA = \frac{3}{\rho_i r_e}$, where $\rho_i$ is the density of ice and $r_e$ is the snow grain size). In our study, wintertime snow grain sizes approach 0.21 mm. The difference is expected as the plateau is generally much colder than Neumayer. The seasonal cycle of modelled average specific surface area in the upper 7 cm (Fig. 8) is comparable to the one presented in Libois et al. (2015), although the wintertime values are probably too low. For the purpose of this study, however, the accurate representation of surface albedo during winter is less relevant as there is no shortwave radiation in winter.

When the adopted albedo values are combined with the observations of SW ↑, the model adequately reproduces the incoming shortwave radiation (Fig. 9, bias = +0.93 W m$^{-2}$, RMSD = 7.3 W m$^{-2}$), providing confidence in the modelled albedo.

## 4.2 Magnitude of the snowmelt–albedo feedback

Three experiments with the SEB model were carried out in addition to the original run ($R_0$), which uses the measured albedo:

- $R_1$: the average measured albedo (0.84, determined by adding all SW ↓ and SW ↑ for all measurements when the sun is higher than 15° above the horizon and taking the ratio between the two) is prescribed for the entire period.

- $R_2$: the full albedo parameterisation is used.

- $R_3$: refrozen snow does not contribute to the changing snow characteristics, i.e. $f_r = 0$ in Eq. (8).

Figure 10a and b show time series of modelled cumulative and seasonal surface melt for the four experiments. Experiment $R_1$ underpredicts melt in most seasons, yielding a mean annual amount of surface melt of 39±27 mm w.e. yr$^{-1}$ (compared to 50±42 mm w.e. yr$^{-1}$ for experiment $R_0$). More melt was modelled in the 1995–1996 melt season, which was characterised by frequent precipitation events and cloudy conditions, keeping observed albedo higher than the long-term mean. Because the albedo parameterisation (used in experiment $R_2$) has been calibrated to match observed albedo, experiment $R_2$ adequately reproduces the amount of seasonal melt (50 ± 34 mm w.e. yr$^{-1}$), although melt, e.g. in the 2012 melt season, is underestimated. Run $R_3$ represents the situation in which the SMAF has been switched off, leading to significantly underpredicted melt (21 ± 16 mm w.e. yr$^{-1}$).

Defining the strength of the SMAF as the ratio between the total seasonal surface melt in experiments $R_2$ and $R_3$, we obtain an average value of 2.6, with a range of 1.3 (1996–1997) to 4.8 (1993–1994; see Fig. 10c). The effect of subsurface penetration of shortwave radiation on this result is esti-

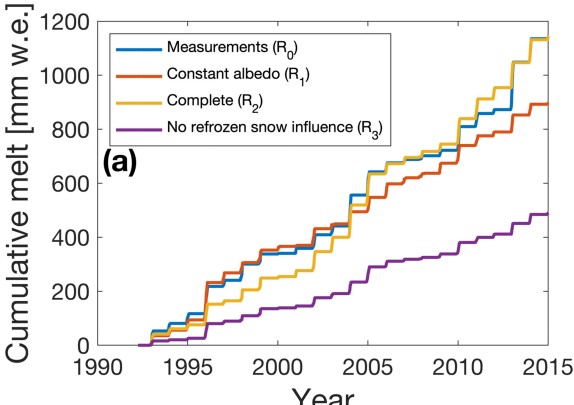

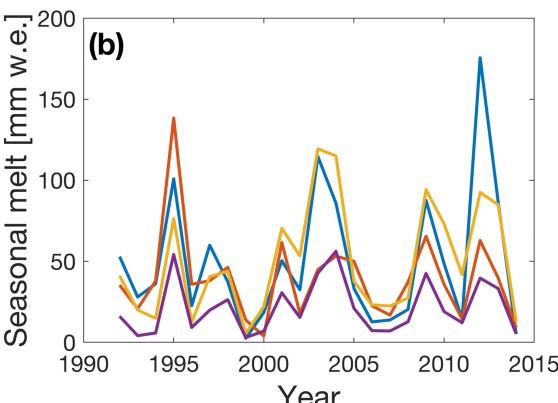

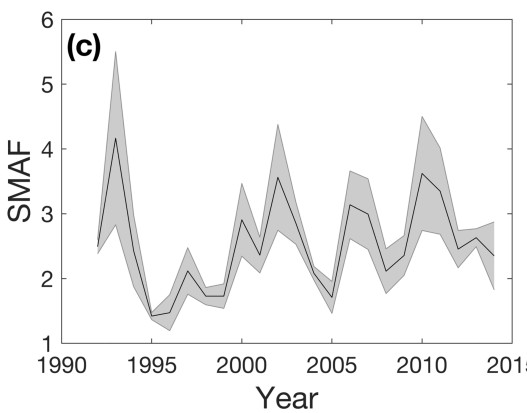

**Figure 10. (a)** Time series of the modelled cumulative amount of melt for the run with measured albedo ($R_0$, blue), a constant albedo of 0.84 ($R_1$, red), a run in which refrozen snow impacts snow grain size ($R_2$, yellow) and a run in which snow grain size is not influenced by refrozen snow ($R_3$, purple). **(b)** Same as panel **(a)** but for seasonal amount of melt. **(c)** Ratio of modelled surface melt between yellow and purple lines in panels **(a)** and **(b)** (runs $R_2$ and $R_3$ respectively). The grey area indicates the uncertainty coming from the uncertainty in the determination of $\tau$ (Fig. 2b), $\pm 5\,\mathrm{W\,m^{-2}}$ measurement uncertainty in SW $\uparrow$ and the inclusion of shortwave radiation penetration.

mated by repeating the above experiments with an inclusion of the radiation penetration model of Kuipers Munneke et al. (2009). This yielded an average SMAF of 2.3, ranging from 1.5 (2005–2006) to 3.2 (2002–2003). The main difference between the two experiments is the reduced interannual variability: including penetration of shortwave radiation does not yield SMAF values larger than 3.5. Shortwave radiation penetration heats the subsurface, causing subsurface melt which is less affected by the SMAF because the radiative flux is smaller in the subsurface. Therefore, the "extreme" years in the sense of SMAF are less distinct in the experiment with shortwave radiation penetration. The effect of shortwave radiation penetration is included in the uncertainties indicated in Fig. 10c. Combining this with the uncertainties in observed SW $\uparrow$ and the determination of $\tau$ (Fig. 2b) leads to uncertainties in the determination of the SMAF of typically 15 %, with a range of 4 % (1995–1996) to 32 % (1993–1994).

A weak positive correlation was found between SMAF and SW $\downarrow$ ($R^2 = 0.15$, $p = 0.07$): if SW $\downarrow$ increases, more energy is available at the surface for melting, which is then in turn further intensified by SMAF. Another weak negative correlation was found between SMAF and summer precipitation ($R^2 = 0.13$, $p = 0.1$): snowfall inhibits SMAF as it effectively "resets" the surface albedo as was also shown by Picard et al. (2012) in a dry region.

Only few studies report on the SMAF concerning the darkening of snow rather than disappearance of it. Box et al. (2012) provide relationships between anomalies of seasonal $T_{2\,\mathrm{m}}$ and SW$_{\mathrm{net}}$ (Figs. 5 and 12 of Box et al., 2012). They find a negative relationship for accumulation regions, i.e. lower 2 m temperatures are associated with smaller SW$_{\mathrm{net}}$. No such relationship is found for Neumayer (not shown).

## 5 Conclusions

In this study, we used 24 years of high-quality meteorological and radiation observations from the BSRN station Neumayer, situated on the Ekström ice shelf, East Antarctica, to force a surface energy balance model. The primary goal was to calculate the amount of melt at Neumayer and to investigate the importance of the snowmelt–albedo feedback (SMAF). Model performance was evaluated based on the difference between modelled and measured surface temperature, and the modelled melt was tested for measurement and model parameter uncertainties. We found that measurement uncertainties, when considered random in time, do not significantly impact modelled melt at Neumayer over the full 24-year period ($< 0.5\,\%$ difference). However, melt amount and model performance are sensitive to the values chosen for the surface roughness length for momentum $z_{0,\mathrm{m}}$ and lower limit of new snow density $\rho_{\mathrm{s,0}}$; thus accurate measurements of these values would further improve future modelling studies. Our results confirm that melt at Neumayer is an intermittent process, occurring on average on only 18 d each sum-

mer, totalling 50 mm w.e. and with an interannual variability of 42 mm w.e. Melt occurs mainly in December and January, sporadically in February and only once melt was modelled in November. Significant and previously unreported trends were found in the net longwave radiation (decreasing) and the sensible heat flux (increasing), but these are unrelated to the melt at Neumayer as they mainly occur in winter and are attributed to a decrease in cloud cover.

The main difference between high and low melt years was found to be surface albedo, implying an important role for the SMAF. We quantified SMAF by implementing and tuning an albedo parameterisation in the SEB model, which includes the effects of snowfall and wet and dry snow metamorphism on albedo. The albedo parameterisation adequately reproduces the seasonal variability in snow grain size, compared to measurements on the Antarctic Plateau (Libois et al., 2015). Our derived wintertime snow grain sizes at Neumayer are somewhat smaller than the satellite-derived summertime snow grain sizes at the Antarctic Plateau (Libois et al., 2015) owing to the lower temperatures on the plateau. Our main finding is that SMAF on average enhances surface melt at Neumayer by a factor of $2.6 \pm 0.8$.

Weak correlations were found of SMAF with summertime SW $\downarrow$ and precipitation ($0.1 < R^2 < 0.2$). To assess how the importance of the snowmelt–albedo feedback varies spatially and temporally, the next step in this research will be applying this method to other sites in Antarctica and a regional climate model (Van Wessem et al., 2018).

*Code and data availability.* The Neumayer data (König-Langlo, 2017) are available upon request via the website of AWI (https://bsrn.awi.de/data/data-retrieval-via-pangaea/, last access: 9 June 2016). The model output is available upon request by the authors.

*Author contributions.* CLJ performed the study and wrote the manuscript. PKM assisted with the implementation of the albedo parameterisation. GKL was in charge of the Neumayer data. CHR, PKM, GKL and MRvdB have commented on the manuscript.

*Competing interests.* The authors declare that they have no conflict of interest.

*Acknowledgements.* We would like to thank the AWI for maintaining the station and the Baseline Surface Radiation Network (BSRN) for providing the data, with special thanks to Amelie Driemel for creating a citation reference and Holger Schmithüsen for helping us to interpret the data. Michiel R. van den Broeke acknowledges support from the Netherlands Earth System Science Centre (NESSC). We would like to thank Ghislain Picard and Achim Heili referee for their constructive comments.

*Financial support.* This research has been supported by the Nederlandse Organisatie voor Wetenschappelijk Onderzoek (grant no. 866.15.204).

*Review statement.* This paper was edited by Mark Flanner and reviewed by Ghislain Picard and Achim Heilig.

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

## Remarks from the language copy-editor

## Remarks from the typesetter