# Peer review of "Quantifying the snowmelt-albedo feedback at Neumayer Station, East Antarctica"

_The Cryosphere, 2018_

## Referee Comment (RC1) · Picard (Referee) · 21 Dec 2018

**Picard (Referee)**

ghislain.picard@univ-grenoble-alpes.fr

Received and published: 21 December 2018

The paper proposed by C.L. Jakobs and colleagues addresses a very important subject that has received little attention for the Antarctic. It aims at demonstrating that snowmelt-albedo feedback is crucial to explain melt dynamics in the coastal Antarctic, which is expected but has never been demonstrated and quantified yet. For this, the paper uses a high-quality and long-term dataset of meteorological conditions from Neumayer station. The dataset is rich enough to allow investigating the surface energy budget in detail and the process underlying the snowmelt-albedo feedback. It is also very long for Antarctic standard (24 years) providing information on long-term changes, with an interesting climate perspective.

The paper is however difficult to read because of the structure, or maybe because some key sentences are missing. The English and style are in contrast excellent. The detailed comments below explain the issues. It is worth noting that they were written while reading the paper for the first time. I decided to keep them as is, despite the fact that some critically missing points were clarified further in the paper. Considering that most readers will read the paper from beginning to end, I think that the order of the comments is helpful to understand the necessary changes. I am optimistic that the authors will solve most of the problems by restructuring the paper and providing the key information early in the paper.

Another issue is the lack of robustness of the results on the feedback with respect to the methodological choices. There are a few questions and suggestions to improve this aspect below, but as a general matter, the paper should be improved by including more comparisons (with results from the literature) and with a proper discussion section putting the results in perspective with respect to other studies having the same aim, but from different regions. Melting snow in the coastal Antarctic is not so different from snow melting in other regions. This should help to consolidate the findings.

Given the great potential of the paper, I encourage the authors to undertake the improvements suggested in the following.

**Detailed comments**

Abstract: the information about the accumulation is missing to my point of view, in order to put in perspective the 46mm w.e., even though there is no direct link for the very specific objective of the study.

In my opinion, using kg/m2 for precipitation and melt is more correct and less confusing than mm w.e.

P2 L13: "Larger snow grains enhance forward scattering of photons". This is a bit incorrect, as it mixes two perspectives (radiative transfer and photon propagation). I
would say "Larger snow grains has reduced scattering relative to absorption" for a pure RT perspective, or "Larger snow grains reduce backward scattering of photons" for a purely photon propagation perspective. This is a detail.

P3L10: The thickness of the top layer is very high, and I suspect that the power of the snowmelt feedback is highly sensitive to this thickness in the range 1-10mm. Skin temperature can also be very different from temperature in the uppermost 4 cm. Using small layers adds complexity which may be inadequate for regional climate modeling, but the scope of the paper is local and process oriented. It is interesting to assess in such conditions how sensitive is the investigated effect to the numerical layer thickness. Tests should be performed to show how robust the results an conclusion are to the thickness of the uppermost layers.

P3L20-25: This simplification is surprising for a study on snowmelt albedo feedback. The effect of the penetration is precisely maximum in the case of coarse/melt grains as the greater absorption is due to a deeper penetration. This seems to me a too extreme simplification given the topic of the paper and past work in this research group. At the minimum this should be assessed, somehow, by a sensitivity analysis. This is also related to the previous comment on numerical layer thickness. The argument about temperature measurement is weak, as measuring temperature in the first centimeters is anyway nearly impossible and secondly because the effect of the penetration (solid greenhouse effect) can be visible in temperature at depth when high quality surface temperature/meteorological conditions are available, as it is the case here.

P3L30: I would remove the emissivity symbol because this equation is only complete for emissivity of 1 (as assumed here). A more correct equation is LWup = sigma eps Ts4 + (1-eps) LWdown. This significantly reduces the sensitivity to eps (as much as the sky is covered by low clouds) compared to the incomplete equation, so would avoid the first part of the comment in P6L26.

P4L5: The approach is surprising, as explained and justified, but I guess this results
from an unsuccessful attempt to, conventionally, use SWdown ? If not, this should obviously be tested. If yes, a more direct explanation of what has been done should be presented with some developments. In particular, a detection and statistical study of riming would be interesting, if this is an important problem to collect the data, in particular on how it correlates with melt (I intuitively expect a negative correlation). This section is confusing.

P4L20: Since grain growth is very sensitive to liquid water content (cubic power) which comes from the melted mass (constrained by available energy) and the layer thickness, this growth is thus very sensitivity to layer thickness (inverse of cubic power?). Here again I suggest to perform a sensitivity to the numerical layer thickness. Exploring the range 1-5cm should be adequate for this aspect, to stay far from divergence at very small layer thicknesses. I'm afraid this sensitivity analysis could greatly affect the result section... and change the paper.

Figure 2: make the individual dots partially transparent (alpha parameter) to better represent the density of dots (or make the dots smaller but the effect is usually better with transparency). The actual representation can be misleading when the number of dots is huge (the case here) and the density is uneven.

Figure 3: The title seems incorrect. Is it right that a sensitivity analysis has been done using a Monte-Carlo approach (chose random pair of z0 and density, run the model and compute RMSE) ? If yes, the graph does not show the relationship between these two parameters, but instead the RMSE and bias as function of both parameters. Still if I understand well, I suggest as a small improvement (for a next paper) to use quasi-random generator instead of pseudo-random. A Voronoi interpolation would also improve the graph. This is not critical.

P6L19-20: This sentence is hard to follow without the formulations. Equations could be added in the method section.

Figure 4: I again suggest transparency on dots + remove non significant numbers for
R2, bias and RMSE (same for Fig 12).

P6L25: Maybe. It could also be a problem of calibration of the radiometer. In such case all the Ts,obs would be scaled down. I suggest to 1) show transparent dots to visualize if these cases are frequent or not, and 2) check that Ts>273.15K occurs mainly for low wind to support the proposed hypothesis of heating. Otherwise, consider to 'recalibrate' the radiometer by scaling down its efficiency to reduce the number of Ts over 273.15K. It may be necessary to use the complete LWup and emissivity close to 0.98 to make this test. Recalibration may lead to a significant effect on snowmelt simulations.

Figure 5: This figure is a bit complex to read despite its apparent simplicity, I have spent some time to understand why the steps and what is the black/red mixture. I suggest to show the grid (vertical dotted gray line on 1st Jan of each year or another way to visualize the summers). The ÂńÂăshaded red areaÂăÂż appears as a line, it would be better to remove it. The necessary info is in the text and is also next to the discussion P6L30-34 which is very good and give a more correct impression of the potential uncertainties than the red area. I'm also wondering about the interest of showing (only) the cumulative melt. I have spent some time to mentally derivate the curve to see the temporal trend and variability (then I realize later it is in Fig 8...). I suggest to add a plot with annual melt along with the cumulative time-series. The measurement error might be more visible on this plot.

Section 3.2. It is relatively disconnected from the remaining. This could be moved to the data section, or at least before Section 3.1

Figure 7: the color is not visible. Is it possible to make wind roses (showing wind speed and direction as e.g. in Champollion et al. 2013 in TC) for 2 or 3 classes of T2m-Ts (e.g. <5 and >5) ? In the end, is the information on temperature so useful ?

P7L19: Is it relative to water or ice ? Relative to ice is more relevant over the ice-sheet.

P7L29: I don't see in Fig 8 and Table 2 that SEB is dominated by SWnet. What does
this mean ? All the plots in Figure 8 have a different y-axis scaling, which makes difficult to judge the dominance of one or another terms.

P8L6: Ts could be shown in Fig 8 (along with Tair).

P8L27: "The difference in SWnet is caused solely by surface albedo". How to exclude the cloudiness as a cause ? Has the LWdown changed between the two years ? More generally how does this interact with the 'unconventional' approach use to compute the SW fluxes. Is it mainly an observational results or an intrinsic consequence of the model and approach ? On a one hand I'm impressed that SW down is equal for both years suggesting that the model predicts the right grain size that perfectly remove the albedo dependence from SWup. However a constant SWdown between both years is only expected if cloudiness has not changed. It is worth checking this, because this is an indirect validation of the approach and of the model grain size.

P9L6: Picard et al. 2012 (doi:10.1038/nclimate1590)Âămay be a useful citation at this point.

P9L17: It is not clear in the data section that SWdown was not excluded (due to riming) and used to compute observe albedo. This Section 4.1 should be moved in the Method section, because it is necessary to understand the previous section results (see my comment P8L27).

P9L25: Picard et al. 2012, Libois et al. 2015 and Picard et al. 2016 provide observed relationship between dry snow albedo and grain size.

P9L27: "to best match the cumulative melt using observed albedo.". I do not understand what has been done. It seems in contradiction with Section 2.2 which indicates that SWdown is not used because unreliable. How to compute valid albedo in these conditions ? In any case this kind of information is required in the method section before the result section Additionally, it seems relevant to show the observed albedo evolution if it exists. TCD
P10L5: CNR4 are given for SZA>60°.

P10L19-20: are these metrics calculated over the summer or the year ?

Section 4.2: From here, I start to understand what I have missed before. It is not clear that the main simulation was done with measurements of SWdown and SWup because the Section 2.2 emphasizes the unconventional approach and the albedo parameterization. I let the previous comments written before reaching this section because they highlight the problem for who reads the paper linearly

Nevertheless, I'm still concerned by the interaction between the approach and the finding of the importance of the snowmelt-albedo feedback. The results seem to entirely rely on the calibration of the metamorphism and albedo parameterizations and their validation is to limited. For instance, over-estimating grain growth in wet conditions automatically leads to over-estimate the importance of SMAF. Ideally, comparison with data from the literature (even on seasonal snow, which is subject to comparable conditions when melting) would help to consolidate a little bit more the result. I was also expecting a discussion section comparing SMAF with the literature.

The discussion at the end of P10 confirms the lack of robustness. The sensitivity to the numerical layer thickness which I propose before is likely to further weaken the findings of this section.

A possible solution is to define SMAF from R0 and R1', where R1' uses the albedo at the end of the winter (and not the annual average of albedo). This would avoid to rely on the grain growth and grain-albedo parameterization, and would be more robust. At least, it should be checked that R1' is close and lower than R3. The main drawback of using R1' is neglecting the dependency on cosine(SZA) which tends to reduce albedo and increase melt during the summer, in parallel with the grain growth.

---

## Referee Comment (RC2) · Anonymous Referee #2 · 27 Dec 2018

The manuscript "Quantifying the snowmelt-albedo feedback at Neumayer Station, East Antarctica" by Jacobs et al. presents meteorological data and simulation results to determine the albedo feedback effect at a single point for an ice shelf region of Antarctica. The chosen location (Neumayer Station) is well-equipped with instruments to measure four component radiation and sensors are maintained regularly. Such data allow for determination of contributing parameters such as surface roughness and microscale wind fields to estimate full energy balance. I consider the quantification of the melt albedo feedback as highly relevant for the cryospheric community especially for snow on ice sheets. However, some missing information as well as the confusing structure of the manuscript prevent publication in the current state. Major points of criticism are:

- The reader gets very confused by the structure of the manuscript. I recommend to

revise carefully. The presented results sections consist of results and discussion, while large fractions of the first results (Section 3) mostly consist of data presentation. In addition, measured data and results simulated by model approaches are constantly mixed in Figures and text. It would be much easier to follow if measured parameters such as temperature, wind, humidity and radiation are separated from generated parameters such as Q\_s, Q\_I etc. Same appears for manuscript sections and paragraphs: for instance, P6 L12-20 is solely discussion same as P6 L29-L3 P7 while before and after those paragraphs you mix measured data and model outputs. In addition, the manuscript title indicates quantification of the melt albedo feedback, while only 2 pages and 2-3 Figures (out of 13 – not mentioning the numerous panels) are referring to snowmelt and albedo feedbacks. I understand that it is necessary to introduce the meteorological data, however, please carefully evaluate the necessity of the presentation of each parameter (Figs 6-9) with sometimes redundancies in the text. Some of the Figures would fit into a supplementary material section. I consider the colorbar in Fig. 7 as being useless. It is impossible to identify differences.

- The nomenclature is sometimes not correct. First of all, what is "fresh snow"? I assume you refer to new snow, which would not be the correct nomenclature either. New snow refers to "Recently fallen snow in which the original form of the ice crystals can be recognized" among others presented in Fierz et al. (2009). The term recently implies a defined time frame. The snow you refer to in the manuscript can rather be defined as near surface snow or surface snow for which you should define a depth range as well. Such a surface snow undergoes rapid transformations especially for polar regions on ice sheets. I am not sure I understand which formulations are used to estimate snow metamorphism at the surface. It might be beyond the scope of the manuscript but you should distinguish between temperature gradient metamorphism (TGM), equi-temperature metamorphism, melt-freeze metamorphism and Firnification and pressure metamorphism. The latter two can be excluded for surface snow but simply assuming grain growth by melt-freeze metamorphism has to be discussed more in detail. Can you present in-situ data on surface densities and grains recorded by the
staff at Neumayer? Please see the following paper for more details on metamorphism (Calonne et al. 2014; doi:10.5194/tc-8-2255-2014). Grain size might be a good tuning parameter but is not a parameter quantifying adequately properties of snow. For the here referred optical properties, it is recommended to use the optical-equivalent grain size or specific surface area (SSA). Again, this might be beyond the scope of the paper but you should at least be up to date with nomenclature and references.

- Please quantify parameterizations (e.g. P9 L16-17).

- Please be consistent: snow pack versus snowpack. I recommend to use snowpack as stated in Fierz et al. 2009. Same appears for T\_s as surface temperature or T\_0 as in Fig. 7 or P3 L10.

Fierz, C., Armstrong, R.L., Durand, Y., Etchevers, P., Greene, E., McClung, D.M., Nishimura, K., Satyawali, P.K. and Sokratov, S.A. 2009. The International Classification for Seasonal Snow on the Ground. IHP-VII Technical Documents in Hydrology N°83, IACS Contribution N°1, UNESCO-IHP, Paris.

---

## Author Comment (AC1) · 11 Mar 2019

Please find the revised manuscript, the revised manuscript with highlighted changes and the responses to both author comments in the supplement.

Please also note the supplement to this comment:
https://www.the-cryosphere-discuss.net/tc-2018-221/tc-2018-221-AC1-supplement.zip

---

## Author Response (AR1)

The paper proposed by C.L. Jakobs and colleagues addresses a very important subject that has received little attention for the Antarctic. It aims at demonstrating that snowmelt-albedo feedback is crucial to explain melt dynamics in the coastal Antarctic, which is expected but has never been demonstrated and quantified yet. For this, the paper uses a high-quality and long-term dataset of meteorological conditions from Neumayer station. The dataset is rich enough to allow investigating the surface energy budget in detail and the process underlying the snowmelt-albedo feedback. It is also very long for Antarctic standard (24 years) providing information on long-term changes, with an interesting climate perspective.

We thank the referee for his kind words. We will comment on each remark below. Text shown here in green has been added to the manuscript.

The paper is however difficult to read because of the structure, or maybe because some key sentences are missing. The English and style are in contrast excellent. The detailed comments below explain the issues. It is worth noting that they were written while reading the paper for the first time. I decided to keep them as is, despite the fact that some critically missing points were clarified further in the paper. Considering that most readers will read the paper from beginning to end, I think that the order of the comments is helpful to understand the necessary changes. I am optimistic that the authors will solve most of the problems by restructuring the paper and providing the key information early in the paper.

We believe that this confusion has mainly arisen from the incorrect suggestion that the albedo parameterisation is used throughout the manuscript rather than only in Sect. 4. This is now mentioned explicitly at multiple locations, which should hopefully improve the structure of the manuscript.

Another issue is the lack of robustness of the results on the feedback with respect to the methodological choices. There are a few questions and suggestions to improve this aspect below, but as a general matter, the paper should be improved by including more comparisons (with results from the literature) and with a proper discussion section putting the results in perspective with respect to other studies having the same aim, but from different regions. Melting snow in the coastal Antarctic is not so different from snow melting in other regions. This should help to consolidate the findings. Given the great potential of the paper, I encourage the authors to undertake the improvements suggested in the following.

We have added comparisons with several studies, focussing on melt climate at Neumayer, grain sizes and the snowmelt-albedo feedback. This is addressed in Sects. 3.4, 4.1 and 4.2. [3.4]

The findings presented in this section are in good agreement with Van den Broeke et al. (2010), who used a similar approach to calculate the SEB at Neumayer, but used a lower value for  $z_{0,m} = 0.32 \text{ mm}$  and a higher snow density that was assumed constant with depth (420 kg m-3 cf. 320 kg m-3). Compared to melt estimates from Larsen C ice shelf, obtained through a similar modelling approach by Kuipers Munneke et al. (2012), melt at Neumayer is weak. Owing to its more northerly location, on Larsen C ice shelf an annual (2009–2011) average melt energy of  $2.8 \text{ W m}^{-2}$  is obtained, compared to the 2009–2011 annual average of  $0.7 \text{ W m}^{-2}$  obtained at Neumayer; furthermore, in November and February melt occurs much more frequently on Larsen C ice shelf.

[4.1]

Libois et al. (2015) and Picard et al. (2016) present observations of snow grain sizes on the Antarctic

plateau during field campaigns in 2012–13 and 2013–14 as well as estimates from satellite observations. On the plateau, summer temperatures are comparable to Neumayer winter temperatures. Libois et al. (2015) report summertime snow grain size estimates of approximately 0.11 mm (Fig. 6 in their study, reported as a specific surface area  $SSA = \frac{3}{\rho_i r_e}$ , where  $\rho_i$  is the density of ice and  $r_e$  is the snow grain size). In our study, wintertime snow grain sizes approach 0.21 mm. The difference is expected as the plateau is generally much colder than Neumayer. The seasonal cycle of modelled average snow grain size in the upper 7 cm (Fig. 8) is comparable to the one presented in Libois et al. (2015).

**[4.2]**

Only few studies report on the snowmelt-albedo feedback concerning the darkening of snow rather than disappearance of it. Box et al. (2012) provide relationships between anomalies of seasonal  $T_{2m}$ and  $SW_{net}$  (Fig. 5 and 12 of Box et al. (2012)). They find a negative relationship for accumulation regions, i.e. lower 2m temperatures are associated with smaller  $SW_{net}$ . No such relationship is found for Neumayer (not shown).

**Detailed comments**

Abstract: the information about the accumulation is missing to my point of view, in order to put in perspective the 46mm w.e., even though there is no direct link for the very specific objective of the study.

We have added to the abstract:

This is a small value compared to an annual average (1992–2016) accumulation of  $415 \pm 86$  mm w.e.

In my opinion, using kg/m2 for precipitation and melt is more correct and less confusing than mm w.e.

To avoid confusion, we have added the following sentence:

Throughout this paper, melt and accumulation amounts are expressed in terms of mm water equivalent (mm w.e.), which equals  $kg m^{-2}$ .

P2 L13: "Larger snow grains enhance forward scattering of photons". This is a bit incorrect, as it mixes two perspectives (radiative transfer and photon propagation). I would say "Larger snow grains has reduced scattering relative to absorption" for a pure RT perspective, or "Larger snow grains reduce backward scattering of photons" for a purely photon propagation perspective. This is a detail.

**Changed**

Larger snow grains reduce backward scattering of photons.

P3L10: The thickness of the top layer is very high, and I suspect that the power of the snowmelt feedback is highly sensitive to this thickness in the range 1-10mm. Skin temperature can also be very different from temperature in the uppermost 4 cm. Using small layers adds complexity which may be inadequate for regional climate modeling, but the scope of the paper is local and process oriented. It is interesting to assess in such conditions how sensitive is the investigated effect to the numerical layer thickness. Tests should be performed to show how robust the results an conclusion are to the thickness of the uppermost layers.

Thank you for this valuable suggestion. We have assessed the impact of snow layer thickness by performing a run with decreased layer thickness, i.e. 1 cm for the top layer instead of 4 cm. Although the simulation without albedo parameterisation showed only a small (< 1%) increase in cumulative

amount of melt, 1154 vs. 1145 mm w.e., we decided to base the reviewed manuscript on the values obtained with the high resolution runs, as we agree that the higher resolution allows for a more accurate calculation of the snow grain size in the upper parts of the snowpack.

P3L20-25: This simplification is surprising for a study on snowmelt albedo feedback. The effect of the penetration is precisely maximum in the case of coarse/melt grains as the greater absorption is due to a deeper penetration. This seems to me a too extreme simplification given the topic of the paper and past work in this research group. At the minimum this should be assessed, somehow, by a sensitivity analysis. This is also related to the previous comment on numerical layer thickness. The argument about temperature measurement is weak, as measuring temperature in the first centimeters is anyway nearly impossible and secondly because the effect of the penetration (solid greenhouse effect) can be visible in temperature at depth when high quality surface temperature/meteorological conditions are available, as it is the case here.

Our initial motivation to neglect shortwave radiation penetration was that we assumed this effect to be small for small-grained Antarctic snow, and that our future aim is to compare SMAF with model results, in which radiation penetration is not yet considered. We however agree with the reviewer that this assumption must be assessed more completely. To take penetration of shortwave radiation into account, we implemented the relatively simple model based on 118 wavelength bands also used by Kuipers Munneke et al. (2009) (doi:10.5194/tc-3-155-2009). It is based on Brandt and Warren (1993) (doi:10.3198/1993JoG39-131-99-110), who used the two-stream model by Schlatter (1972) (doi:10.1175/1520-0450(1972)011<1048:TLSEBA>2.0.CO;2). In this model, the amount of absorbed shortwave radiation amongst other things depends on layer density and layer grain size, which are provided in look-up tables for seven different snow grain sizes, which we interpolate to the grain sizes obtained from the albedo parameterisation.

A melt increase of 13% is found when radiation penetration is included: 1326 mm w.e. compared to 1154 mm w.e., but average SMAF did not change. We thus conclude that including penetration of shortwave radiation using a simple radiation transport model does increase the amount of modelled melt in an absolute sense, but that the SMAF results are robust to both the layer thickness and whether or not shortwave radiation penetration is included. We now include a discussion on the effect of shortwave radiation penetration (see below), and included the effect in the SMAF uncertainty estimate in Fig. 14.

[2.1]

Subsurface penetration of shortwave radiation is calculated using a spectral model (Kuipers Munneke et al., 2009), based on the parameterisation by Brandt and Warren (1993), which is in turn based on the two-stream radiation model of Schlatter (1972). The impact on modelled melt and the quantification of the snowmelt-albedo feedback is discussed in the relevant sections.

**[3.3]**

Using the subsurface radiation model of Kuipers Munneke et al. (2009), the influence of subsurface penetration of shortwave radiation is estimated. Its inclusion increases the cumulative amount of melt by 13%, from 1154 mm w.e. to 1326 mm w.e. The absorbed shortwave radiation heats the subsurface layers, but the heat cannot be transported away as effectively as would happen at the surface by turbulent fluxes and longwave radiation. This leads to an increase in total melt.

**[4.2]**

The effect of subsurface penetration of shortwave radiation on this result is estimated by repeating

the above experiments with inclusion of the radiation penetration model of Kuipers Munneke et al. (2009). This yielded an average SMAF of 2.3, ranging from 1.5 (2005–06) to 3.2 (2002–03). The main difference between the two experiments is the reduced interannual variability: including penetration of shortwave radiation does not yield SMAF values larger than 3.5. Shortwave radiation penetration heats the subsurface, causing subsurface melt which is less affected by the snowmelt-albedo feedback because the radiative flux is smaller in the subsurface. Therefore, the 'extreme' years in the sense of SMAF are less distinct in the experiment with shortwave radiation penetration. The effect of shortwave radiation penetration is included in the uncertainties indicated in Fig. 10c. Combining this with the uncertainties in observed  $SW \uparrow$  and the determination of  $\tau$  (Fig. 2b) leads to uncertainties in the determination of the SMAF of typically 15%, with a range of 4% (1995–96) to 32% (1993–94).

P3L30: I would remove the emissivity symbol because this equation is only complete for emissivity of 1 (as assumed here). A more correct equation is  $LWup = sigma eps Ts^4 + (1-eps) LWdown$ . This significantly reduces the sensitivity to eps (as much as the sky is covered by low clouds) compared to the incomplete equation, so would avoid the first part of the comment in P6L26.

The model is able to work with an emissivity different from 1 and in that case it will employ the correct equation. As in this study  $\epsilon = 1$ , it is not necessary to write it in the equation. We have now clarified this.

(...) using Stefan-Boltzmann's law for a longwave emissivity  $\epsilon = 1$ :

$$LW \uparrow = \sigma T_s^4, \tag{2}$$

P4L5: The approach is surprising, as explained and justified, but I guess this results from an unsuccessful attempt to, conventionally, use SWdown ? If not, this should obviously be tested. If yes, a more direct explanation of what has been done should be presented with some developments. In particular, a detection and statistical study of riming would be interesting, if this is an important problem to collect the data, in particular on how it correlates with melt (I intuitively expect a negative correlation). This section is confusing.

The choice to use measured  $SW \uparrow$  instead of measured  $SW \downarrow$  (and 'calculating'  $SW \downarrow$ ) is motivated by findings by e.g. Van den Broeke et al. (2004) and Smeets et al. (2018) who showed that the upward-facing sensor (which measures an important direct radiation component) is more prone to inaccuracies due to tilt and riming. The simulations with the observed albedo use  $SW \uparrow$  combined with the 24-hour running mean albedo to construct  $SW \downarrow$  and  $SW_{net}$ . Using a running mean albedo instead of the instantaneous observed albedo further reduces measurement errors due to tilt and riming. The simulations with parameterised albedo use  $SW \uparrow$  combined with the parameterised albedo to obtain  $SW \downarrow$  and  $SW_{net}$ . We now cite the paper in which this method is explained (Van den Broeke et al., 2004), and added to the manuscript:

Because the shortwave radiation sensor faces the sky and includes a significant direct component, measured  $SW \downarrow$  suffers from relatively large uncertainties owing to poor sensor cosine response, sensor tilt and/or rime formation (Smeets et al., 2018). In order to improve the accuracy of observed net shortwave radiation used in the SEB calculations (Sect. 3), we calculate  $SW_{net}$  based on  $SW \uparrow$ , which is diffuse and hence much less sensitive to these errors, in combination with a 24-hour moving average albedo, as described in Van den Broeke et al. (2004). In Sect. 4, in which albedo is parameterised to study melt-albedo feedbacks, for consistency we use measured  $SW \uparrow$  in combination with parameterised albedo to estimate  $SW_{net}$ . P4L20: Since grain growth is very sensitive to liquid water content (cubic power) which comes from the melted mass (constrained by available energy) and the layer thickness, this growth is thus very sensitivity to layer thickness (inverse of cubic power?). Here again I suggest to perform a sensitivity to the numerical layer thickness. Exploring the range 1-5cm should be adequate for this aspect, to stay far from divergence at very small layer thicknesses. I'm afraid this sensitivity analysis could greatly affect the result section... and change the paper.

The effect of upper layer thickness was addressed in the response to the comment on P3L10. Only small differences were found, and the current values in the manuscript are obtained from simulations with a smaller layer thickness (1 cm).

Figure 2: make the individual dots partially transparent (alpha parameter) to better represent the density of dots (or make the dots smaller but the effect is usually better with transparency). The actual representation can be misleading when the number of dots is huge (the case here) and the density is uneven.

Good suggestion, we have changed the figures accordingly.

Figure 3: The title seems incorrect. Is it right that a sensitivity analysis has been done using a Monte-Carlo approach (chose random pair of z0 and density, run the model and compute RMSE) ? If yes, the graph does not show the relationship between these two parameters, but instead the RMSE and bias as function of both parameters. Still if I understand well, I suggest as a small improvement (for a next paper) to use quasi-random generator instead of pseudo-random. A Voronoi interpolation would also improve the graph. This is not critical.

Yes, a Monte-Carlo approach was used for this sensitivity analysis. However, we decided to remove Fig. 3 as it is not important for the final experiments and conclusions.

P6L19-20: This sentence is hard to follow without the formulations. Equations could be added in the method section.

We have added the relevant formulations in the methods section.

Surface roughness lengths for momentum, heat and moisture are related through the expression of Andreas (1987):

$$\ln\left(\frac{z_{0*}}{z_{0,m}}\right) = a_1 + a_2 \ln\left(\text{Re}_*\right) + a_3 \ln\left(\text{Re}_*\right)^2,\tag{3}$$

where  $z_{0*}$  represents either  $z_{0,h}$  or  $z_{0,q}$ , the roughness lengths for heat and moisture respectively,  $a_1, a_2$  and  $a_3$  are coefficients determined by Andreas (1987) for various regimes of the roughness Reynolds number  $\operatorname{Re}_* = \frac{u_* z_{0,m}}{\nu}$  with kinematic viscosity  $\nu$  and friction velocity  $u_*$ .

Figure 4: I again suggest transparency on dots + remove non significant numbers for R2, bias and RMSE (same for Fig 12).

We have changed the figure, removed the digits.

P6L25: Maybe. It could also be a problem of calibration of the radiometer. In such case all the Ts,obs would be scaled down. I suggest to 1) show transparent dots to visualize if these cases are frequent or not, and 2) check that Ts>273.15K occurs mainly for low wind to support the proposed hypothesis of heating. Otherwise, consider to 'recalibrate' the radiometer by scaling down its efficiency to reduce the number of Ts over 273.15K. It may be necessary to use the complete LWup and emissivity close to 0.98 to make this test. Recalibration may lead to a significant effect on snowmelt simulations.

Measured  $T_s > 273.15$  K values were only present in the first couple of years, afterwards they were removed through post-processing by AWI.  $T_S > 273.15$  K indeed occurred mostly when wind speed was relatively low (potentially causing heating of sensor window). Furthermore, measurements of  $T_s > 273.15$  K could be a result of the radiation sensor partly measuring longwave radiation emitted by the air between the surface and the sensor at 2 metre height, but we deem this less likely in this cold environment. We added to the manuscript:

Measured values of  $T_s$  in excess of the melting point in Fig. 4 only occurred in the first 6 seasons; from 1998–99 onwards they were removed by additional post-processing. These measurements mainly reflect uncertainties in the adopted unit value of longwave emissivity and in measured  $LW \uparrow$ , e.g. from sensor window heating (Smeets et al., 2018) and the fact that the downward facing radiation sensor also measures longwave radiation emitted by the relatively warm air between the surface and the sensor.

Figure 5: This figure is a bit complex to read despite its apparent simplicity, I have spent some time to understand why the steps and what is the black/red mixture. I suggest to show the grid (vertical dotted gray line on 1st Jan of each year or another way to visualize the summers). The "shaded red area" appears as a line, it would be better to remove it. The necessary info is in the text and is also next to the discussion P6L30-34 which is very good and give a more correct impression of the potential uncertainties than the red area. I'm also wondering about the interest of showing (only) the cumulative melt. I have spent some time to mentally derivate the curve to see the temporal trend and variability (then I realize later it is in Fig 8...). I suggest to add a plot with annual melt along with the cumulative time-series. The measurement error might be more visible on this plot. We have added grey patches from 1 Nov–1 Mar to help the reader in finding the melt seasons. The shaded red area indeed is very narrow, owing to the small uncertainty due to the measurement errors, so we have removed it. We have added a second plot with seasonal melt amounts, also without the uncertainty due to measurement errors as it was barely visible anyway. We changed the caption accordingly:

Effect of model uncertainties on (a) cumulative melt and (b) seasonal melt. The shaded area indicates the  $1\sigma$  range due to model uncertainties (changing  $z_{0,m}$  and  $\rho_{s,0}$  between their respective values) which is asymmetrical because the values that are used for the rest of the study ( $z_{0,m} =$  $1.65 \text{ mm}, \rho_{s,0} = 280 \text{ kg m}^{-3}$ ) are not in the middle of the range that was probed. The vertical grey patches in (a) indicate Nov-Feb of each year. Note that (b) ends earlier than (a) because the observations do not cover the 2015–16 melt season entirely.

Section 3.2. It is relatively disconnected from the remaining. This could be moved to the data section, or at least before Section 3.1

We have moved the discussion about the local climate to Sect. 2.3.1.

Figure 7: the color is not visible. Is it possible to make wind roses (showing wind speed and direction as e.g. in Champollion et al. 2013 in TC) for 2 or 3 classes of T2m-Ts (e.g. <5 and >5)? In the end, is the information on temperature so useful ?

We agree that the discussion about wind direction and wind speed does not contribute to the paper final discussion and conclusions. We have therefore removed this figure.

P7L19: Is it relative to water or ice? Relative to ice is more relevant over the ice-sheet.

The conversion from specific humidity to relative humidity takes the present air temperature into account. Depending on the prevailing air temperature, the saturated vapour pressure with respect

**to either water or ice is used. We added:**

(relative to either water or ice, depending on the air temperature)

P7L29: I don't see in Fig 8 and Table 2 that SEB is dominated by SWnet. What does this mean ? All the plots in Figure 8 have a different y-axis scaling, which makes difficult to judge the dominance of one or another terms.

The reference should have been Fig. 6b (now Fig. 3b), which presents the seasonal cycle of the SEB components, and, as we should have stressed, *in summer*. This has now been corrected

Annual (Mar–Feb) mean values of near-surface meteorological quantities and SEB components are presented in Table 2, with seasonal cycles of SEB components presented in Fig. 3b. These show that the summertime SEB is dominated by the radiation fluxes...

P8L6: Ts could be shown in Fig 8 (along with Tair). We have removed the panel with the timeseries as we think they do not contribute to the overall goal of the study.

P8L27: "The difference in SWnet is caused solely by surface albedo". How to exclude the cloudiness as a cause ? Has the LWdown changed between the two years ? More generally how does this interact with the 'unconventional' approach use to compute the SW fluxes. Is it mainly an observational results or an intrinsic consequence of the model and approach ? On a one hand I'm impressed that SW down is equal for both years suggesting that the model predicts the right grain size that perfectly remove the albedo dependence from SWup. However a constant SWdown between both years is only expected if cloudiness has not changed. It is worth checking this, because this is an indirect validation of the approach and of the model grain size.

From the original manuscript it was unclear that these simulations have been performed with the observed albedo instead of the parameterised albedo. Therefore, there is no 'prediction' for grain size by the model in this figure. We have included more components in Fig. 10 to show that both  $SW \downarrow$  and  $LW \downarrow$  do not differ between these years. Therefore, we conclude that the difference in  $SW_{net}$  comes from  $SW \uparrow$ , driven by changes in surface albedo. In response,  $LW \uparrow$  changes as the surface is warmer in high melt years, leading to the change in  $LW_{net}$ . We added to the manuscript:  $SW \downarrow$  and  $LW \downarrow$  show almost no difference between high and low melt seasons; therefore, the difference in  $SW_{net}$  cannot be caused by a change in cloud cover and is likely caused solely by surface albedo...

P9L6: Picard et al. 2012 (doi:10.1038/nclimate1590) may be a useful citation at this point. We have added this citation.

Precipitation of new, fine-grained snow has been shown to inhibit the albedo decrease by metamorphism on the Antarctic plateau (Picard et al., 2012).

P9L17: It is not clear in the data section that SWdown was not excluded (due to riming) and used to compute observe albedo. This Section 4.1 should be moved in the Method section, because it is necessary to understand the previous section results (see my comment P8L27).

Thank you for your suggestion. This is achieved by restructuring and adding sentences to clarify the difference between the simulations for Sects. 3 and 4 (see response to second comment).

P9L25: Picard et al. 2012, Libois et al. 2015 and Picard et al. 2016 provide observed relationship

between dry snow albedo and grain size.

We have added a comparison of snow grain sizes in the manuscript, as mentioned in the response to the third general comment.

Libois et al. (2015) and Picard et al. (2016) present observations of snow grain sizes on the Antarctic plateau during field campaigns in 2012–13 and 2013–14 as well as estimates from satellite observations. On the plateau, summer temperatures are comparable to Neumayer winter temperatures. Libois et al. (2015) report summertime snow grain size estimates of approximately 0.11 mm (Fig. 6 in their study, reported as a specific surface area  $SSA = \frac{3}{\rho_i r_e}$ , where  $\rho_i$  is the density of ice and  $r_e$  is the snow grain size). In our study, wintertime snow grain sizes approach 0.21 mm. The difference is expected as the plateau is generally much colder than Neumayer. The seasonal cycle of modelled average snow grain size in the upper 7 cm (Fig. 8) is comparable to the one presented in Libois et al. (2015).

P9L27: "to best match the cumulative melt using observed albedo.". I do not understand what has been done. It seems in contradiction with Section 2.2 which indicates that SWdown is not used because unreliable. How to compute valid albedo in these conditions? In any case this kind of information is required in the method section before the result section Additionally, it seems relevant to show the observed albedo evolution if it exists.

By explicitly stating that the observed albedo was used in Sect. 3 we believe it is now clear what has been done to have the simulation with parameterised albedo adequately reproduce the cumulative melt of the simulation with observed albedo.

**P10L5: CNR4 are given for SZA>60.**

At Neumayer, not the rather simple CNR4-net radiometer was used but more sophisticated pyranometers. In 1992 the AWI started using artificially ventilated K&Z CM11 instruments and in 2009 switched to the even better K&Z CM22. Both have a much better cosine response compared to the CNR4. The cosine error for solar zenith angles greater than 60 degrees for the used pyranometers is part of the measurement errors listed in Table 1.

P10L19-20: are these metrics calculated over the summer or the year ?

Only summer values are used for these metrics.

A weak positive correlation was found between SMAF and  $SW \downarrow (R^2 = 0.15, p = 0.07)$ ; if  $SW \downarrow$  increases, more energy is available at the surface for melting, which is then in turn further intensified by SMAF. Another weak negative correlation was found between SMAF and summer precipitation  $(R^2 = 0.13, p = 0.09)$ ; snowfall inhibits SMAF as it effectively 'resets' the surface albedo as was also shown by Picard et al. (2012).

Section 4.2: From here, I start to understand what I have missed before. It is not clear that the main simulation was done with measurements of SWdown and SWup because the Section 2.2 emphasizes the unconventional approach and the albedo parameterization. I let the previous comments written before reaching this section because they highlight the problem for who reads the paper linearly Nevertheless, I'm still concerned by the interaction between the approach and the finding of the importance of the snowmelt-albedo feedback. The results seem to entirely rely on the calibration of the metamorphism and albedo parameterizations and their validation is to limited. For instance, over-estimating grain growth in wet conditions automatically leads to over-estimate the importance of SMAF. Ideally, comparison with data from the literature (even on seasonal snow, which is subject to comparable conditions when melting) would help to consolidate a little bit more the result. I was also expecting a discussion section comparing SMAF with the literature.

As mentioned in response to the third general comment, we changed the structure of the paper and added sentences in such a way that it is now more clear what has been done to obtain the optimal settings for grain size calculation and the associated parameterised albedo. Furthermore, we included comparisons with several other studies, in Sects. 3.4, 4.1 and 4.2.

The discussion at the end of P10 confirms the lack of robustness. The sensitivity to the numerical layer thickness which I propose before is likely to further weaken the findings of this section.

We now included several comparisons with literature, as mentioned in the response to the third comment. We also now more clearly emphasise the use of observed albedo in Sect. 3 and the parameterised albedo in Sect. 4, such that it is immediately clear that the results presented in Sect. 3 do not rely on the albedo parameterisation.

A possible solution is to define SMAF from R0 and R1', where R1' uses the albedo at the end of the winter (and not the annual average of albedo). This would avoid to rely on the grain growth and grain-albedo parameterization, and would be more robust. At least, it should be checked that R1' is close and lower than R3. The main drawback of using R1' is neglecting the dependency on cosine(SZA) which tends to reduce albedo and increase melt during the summer, in parallel with the grain growth.

The average albedo at the end of the winter (taken as the first day that the Sun rises above  $10^{\circ}$  altitude, and then take the preceding 48-hour mean albedo) is 0.87 (cf. the full period average albedo of 0.84). Prescribing this albedo throughout the run yields a cumulative amount of melt of 460 mm w.e., and subsequently a SMAF of 2.5 (slightly higher than the high resolution run, which yielded a SMAF of 2.4). The total amount of melt in the  $R'_1$  run is slightly higher than that was modelled by  $R_3$  (which totalled 428 mm w.e.). The variability in SMAF according to  $R'_1$  is much larger than the one calculated by  $R_3$  and sometimes becomes less than one, which is unphysical. As pointed out by the referee, using this measure neglects the dependency on the zenith angle and the impact of precipitation or the periods between precipitation events during the summer season. Therefore, we decided to keep the original definition of SMAF in the manuscript. We have added: Alternatively, SMAF could be defined as the ratio between  $R_0$  and  $R_3$ , or the ratio between  $R_0$  and  $R'_1$ , where  $R'_1$  uses the average albedo at the end of the winter. Using the former definition, the results become more prone to noise due to the performance of the albedo parameterisation itself. The latter definition neglects the dependency on solar zenith angle and the impact of precipitation. Therefore, we believe defining SMAF as the ratio between  $R_2$  and  $R_3$  is more consistent.

The manuscript "Quantifying the snowmelt-albedo feedback at Neumayer Station, East Antarctica" by Jacobs et al. presents meteorological data and simulation results to determine the albedo feedback effect at a single point for an ice shelf region of Antarctica. The chosen location (Neumayer Station) is well-equipped with instruments to measure four component radiation and sensors are maintained regularly. Such data allow for determination of contributing parameters such as surface roughness and microscale wind fields to estimate full energy balance. I consider the quantification of the melt albedo feedback as highly relevant for the cryospheric community especially for snow on ice sheets. However, some missing information as well as the confusing structure of the manuscript prevent publication in the current state.

We thank the referee for their constructive comments. They are addressed below in a structured way. Text in green shows text as it is now in the manuscript.

Major points of criticism are:

- The reader gets very confused by the structure of the manuscript. I recommend to revise carefully. The presented results sections consist of results and discussion, while large fractions of the first results (Section 3) mostly consist of data presentation. In addition, measured data and results simulated by model approaches are constantly mixed in Figures and text. It would be much easier to follow if measured parameters such as temperature, wind, humidity and radiation are separated from generated parameters such as  $Q_s$ ,  $Q_l$  etc.

Thank you for this suggestion. We have separated the presentation of the local near-surface climate from the discussion of the surface energy balance. The near-surface climate discussion is moved to Sect. 2.

Same appears for manuscript sections and paragraphs: for instance, P6 L12-20 is solely discussion same as P6 L29-L3 P7 while before and after those paragraphs you mix measured data and model outputs.

We have moved blocks of text in Sect. 3.2 in order to present and discuss the results in a more logical manner.

In addition, the manuscript title indicates quantification of the melt albedo feedback, while only 2 pages and 2-3 Figures (out of 13 – not mentioning the numerous panels) are referring to snowmelt and albedo feedbacks. I understand that it is necessary to introduce the meteorological data, however, please carefully evaluate the necessity of the presentation of each parameter (Figs 6-9) with sometimes redundancies in the text. Some of the Figures would fit into a supplementary material section. I consider the colorbar in Fig. 7 as being useless. It is impossible to identify differences.

We have removed Figs. 3, 7, 8 and 11 to enhance readability.

- The nomenclature is sometimes not correct. First of all, what is "fresh snow"? I assume you refer to new snow, which would not be the correct nomenclature either. New snow refers to "Recently fallen snow in which the original form of the ice crystals can be recognized" among others presented in Fierz et al. (2009). The term recently implies a defined time frame. The snow you refer to in the manuscript can rather be defined as near surface snow or surface snow for which you should define a depth range as well. Such a surface snow undergoes rapid transformations especially for polar regions on ice sheets.

With fresh snow we refer to new snow as defined by Fierz et al. (2009). The contribution of "fresh

snow" as it is in the albedo parameterisation solely comes from recently fallen snow in a defined time frame, namely the timestep of the model. Snow that was already present in the layer from the previous timestep is considered "old snow", which undergoes the dry snow metamorphism. We have now made this more clear in the manuscript by changing "fresh snow" to "new snow" throughout the manuscript.

I am not sure I understand which formulations are used to estimate snow metamorphism at the surface. It might be beyond the scope of the manuscript but you should distinguish between temperature gradient metamorphism (TGM), equi-temperature metamorphism, melt-freeze metamorphism and Firnification and pressure metamorphism. The latter two can be excluded for surface snow but simply assuming grain growth by melt-freeze metamorphism has to be discussed more in detail. Can you present in-situ data on surface densities and grains recorded by the staff at Neumayer? Please see the following paper for more details on metamorphism (Calonne et al. 2014;doi:10.5194/tc-8-2255-2014). Grain size might be a good tuning parameter but is not a parameter quantifying adequately properties of snow. For the here referred optical properties, it is recommended to use the optical-equivalent grain size or specific surface area (SSA). Again, this might be beyond the scope of the paper but you should at least be up to date with nomenclature and references.

We have added the formulation of dry snow metamorphism. Unfortunately, no in-situ data on surface densities or grains are available from Neumayer. We have now included a comparison with measurements on the Antarctic Plateau (Picard et al. 2012; doi:10.1038/nclimate1590). Although the local climates are very different, the comparison shows that the grain sizes measured on the plateau in summer are similar to the grain sizes modelled at Neumayer in winter. Neumayer winter temperatures are somewhat comparable to plateau summer temperature. We added to the manuscript:

[2.2]

Dry snow metamorphism is parameterised following Kuipers Munneke et al. (2011b):

$$\frac{\mathrm{d}r_{e,dry}}{\mathrm{d}t} = \left(\frac{\mathrm{d}r_e}{\mathrm{d}t}\right)_0 \left(\frac{\eta}{(r_e - r_{e,0}) + \eta}\right)^{1/\kappa},\tag{9}$$

where  $r_{e,0}$  is the new snow grain size, and the coefficients  $\left(\frac{\mathrm{d}r_e}{\mathrm{d}t}\right)_0$ ,  $\eta$  and  $\kappa$  are obtained from a look-up table. This look-up table is compiled based on simulations with the SNICAR model (Flanner et al., 2006), which calculates the snow metamorphism resulting from temperature gradient metamorphism.

[4.1]

Libois et al. (2015) and Picard et al. (2016) present observations of snow grain sizes on the Antarctic plateau during field campaigns in 2012–13 and 2013–14 as well as estimates from satellite observations. On the plateau, summer temperatures are comparable to Neumayer winter temperatures. Libois et al. (2015) report summertime snow grain size estimates of approximately 0.11 mm (Fig. 6 in their study, reported as a specific surface area  $SSA = \frac{3}{\rho_i r_e}$ , where  $\rho_i$  is the density of ice and  $r_e$  is the snow grain size). In our study, wintertime snow grain sizes approach 0.21 mm. The difference is expected as the plateau is generally much colder than Neumayer. The seasonal cycle of modelled average snow grain size in the upper 7 cm (Fig. 8) is comparable to the one presented in Libois et al. (2015).

- Please quantify parameterizations (e.g. P9 L16-17).

We have added the ranges that were probed for the new snow and refrozen snow grain sizes. These parameters were varied within reasonable ranges to optimise the results: new snow grain size between 0.04 mm and 0.3 mm, refrozen snow grain size between 0.1 mm and 10 mm.

- Please be consistent: snow pack versus snowpack. I recommend to use snowpack as stated in Fierz et al. 2009. Same appears for  $T_s$  as surface temperature or  $T_0$  as in Fig. 7 or P3 L10. We have changed snow pack to snowpack throughout the manuscript.  $T_0$  on P3 L10 should have been a  $T_s$ .

**Quantifying the snowmelt-albedo feedback at Neumayer Station, East Antarctica**

Constantijn L. Jakobs1, Carleen H. Reijmer1, Peter Kuipers Munneke1, Gert König-Langlo2, and Michiel R. van den Broeke1

[revised manuscript text omitted]

---

## Author Response (AR2)

We would like to thank the referee for his constructive comments on the revised manuscript. Please find our comments to each remark below. Text shown in green has been added to the manuscript.

P2L15: add "only"
We have added this word:
...snow that has been subjected to only dry compaction...

P3L2: remove "briefly"
We have removed the word.

P4L13: 'For this, we assume albedo is equal to a 24-hour averaging of measured albedo'
We have split the sentence into two sentences and we have added:
To further decrease the impact of these errors, we use a 24-hour moving average albedo, as described in Van den Broeke et al. (2004).

P7L19: This sentence is hard to follow
We have split this sentence into two sentences and we have rewritten part of it:
This also explains why the model outcome is much more sensitive to different values of $z_{0,m}$, as these runs effectively introduce a systematic error between the true (unknown) value and the chosen value. Furthermore, this approach assumes the true value to be constant, which likely is an oversimplification (Smeets and Van den Broeke, 2008).

P7L22: add "of the cumulative melt"
We have added this:
The sensitivity of modelled cumulative melt to $z_{0,m}$ is somewhat unexpected

P7L23: Overall this part is very difficult to understand. Better reference to the figure and explicit value would maybe help.
We have added some text and a citation to the model description (Sect. 2.1). We believe this clarifies the reasoning in this part.
[2.1]
Turbulent fluxes are calculated following the 'bulk' method, which is based on Monin-Obukhov similarity theory (see e.g. Van den Broeke et al. (2006) for relevant equations) between a single measurement level (2 m for temperature and humidity, 10 m for wind) and the surface, assuming the latter to be saturated with respect to ice and using the stability functions according to Dyer (1974) for unstable and Holtslag and De Bruin (1988) for stable conditions.

[3.1]
The sensitivity of modelled cumulative melt to $z_{0,m}$ is somewhat unexpected. Following Eq. (3) both $z_{0,h}$ and $z_{0,q}$ decrease for increasing $z_{0,m}$; in combination with the bulk method this acts to dampen the effect of $z_{0,m}$ on the magnitude of the turbulent fluxes. Our interpretation of this result is that decreasing $z_{0,m}$ and $\rho_{s,0}$ leads to a decrease of the turbulent fluxes as well as the ground heat flux $Q_G$. This reduces the efficiency with which heat is removed from the surface, in turn allowing more energy to be invested in melt.

P7L25: value of what?
This should have been the obtained value **of** $z_{0,m}$.

P7L27: Should it be a different paragraph. What is the link with the previous sentence ?
Yes, this should be a new paragraph.

P8L6+L7: 'Statistically significant'. Because I don't believe there are physically significant [trends] [previously unreported trends] over the 24 years (check)
Remove 'both a result of wintertime trends' or make a second sentence
Corrected, this part now reads:
Statistically significant and previously unreported trends over the full 24-year period (not shown) are detected in $LW \uparrow$ $(-0.28 \pm 0.14\,\mathrm{W\,m^{-2}\,yr^{-1}})$ and $Q_S$ $(+0.21 \pm 0.07\,\mathrm{W\,m^{-2}\,yr^{-1}})$. Both of these are a result of wintertime trends. $LW \uparrow$ is linked directly to $T_s$, which shows a statistically insignificant negative trend

P8L9: which distribution is used?
We assumed normal distribution for both trends. We have added:
. . . which in magnitude exceeds the negative trend in $T_{2m}$ $(-0.0045 \pm 0.02\,\mathrm{K\,yr^{-1}}$; assuming a normal distribution, the probability that the negative trend in $T_s$ is greater in magnitude than the trend in $T_{2m}$ is 0.76).

P8L9: add "air"
Corrected.

P9L7: "$420\,\mathrm{kg\,m^{-3}}$ cf. $320\,\mathrm{kg\,m^{-3}}$" ???
320 should have been 280, this is now corrected. It is a comparison of the value of snow density used by Van den Broeke et al. (2010) and the one used in this study.
$420\,\mathrm{kg\,m^{-3}}$ in their study versus $280\,\mathrm{kg\,m^{-3}}$ in this study

P10L26: Figures
Corrected.

P10L27: vs. ?
We changed it to "compared to".
. . . yielding a mean annual amount of surface melt of $39 \pm 27\,\mathrm{mm\,w.e.\,yr^{-1}}$ (compared to $50 \pm 42\,\mathrm{mm\,w.e.\,yr^{-1}}$ for experiment $R_0$).

P10L30: R2 has been calibrated to do so. Right ? The sentence should make this clear. This does not impact the conclusion.
Correct. We have added some text to make this clear:
Because the albedo parameterisation (used in experiment $R_2$) has been calibrated to match observed albedo, experiment $R_2$ adequately reproduces. . .

P11L15: add "in a dry region"
We have added this.

P11L15: I think it is not necessary in the main document as it adds more noise in a already-difficult-to-follow section. Your comment in the review is sufficient.
We have removed from "Alternatively, SMAF could. . ." until ". . . is more consistent".

P11L30: add "over the period of 24 years"
We have added:
...modelled melt at Neumayer over the full 24-year period ($<$0.5 % difference).

P22: "which is asymmetrical...that was probed" I would remove this. Hard to follow
We have removed this.

P25: add the years here
Corrected.

P26: I'm very skeptical with the highest values. This is far too low compared to what we observe, even in the Alps. I would gray the winter part for which your model is not optimal.
Similar to Fig. 5a we decided to shade the summer months to indicate the relevant parts of the year. We have added to Sect. 4.1:

[revised manuscript text omitted]